# Universal selective transfer printing via micro-vacuum force

Sang Hyun Park[1,5], Tae Jin Kim[1,5], Han Eol Lee [2,5], Boo Soo Ma[3], Myoung Song [3], Min Seo Kim[1], Jung Ho Shin[1], Seung Hyung Lee[1], Jae Hee Lee[1], Young Bin Kim[1], Ki Yun Nam[1], Hong-Jin Park[4], Taek-Soo Kim[3] & Keon Jae Lee [1] ✉

Transfer printing of inorganic thin-film semiconductors has attracted considerable attention to realize high-performance soft electronics on unusual substrates. However, conventional transfer technologies including elastomeric transfer printing, laser-assisted transfer, and electrostatic transfer still have challenging issues such as stamp reusability, additional adhesives, and device damage. Here, a micro-vacuum assisted selective transfer is reported to assemble micro-sized inorganic semiconductors onto unconventional substrates. 20 μm-sized micro-hole arrays are formed via laser-induced etching technology on a glass substrate. The vacuum controllable module, consisting of a laser-drilled glass and hard-polydimethylsiloxane micro-channels, enables selective modulation of micro-vacuum suction force on microchip arrays. Ultrahigh adhesion switchability of $3.364 \times 10^6$, accomplished by pressure control during the micro-vacuum transfer procedure, facilitates the pick-up and release of thin-film semiconductors without additional adhesives and chip damage. Heterogeneous integration of III-V materials and silicon is demonstrated by assembling microchips with diverse shapes and sizes from different mother wafers on the same plane. Multiple selective transfers are implemented by independent pressure control of two separate vacuum channels with a high transfer yield of 98.06%. Finally, flexible micro light-emitting diodes and transistors with uniform electrical/optical properties are fabricated via micro-vacuum assisted selective transfer.

In the upcoming era of the Internet of Things (IoT), soft electronics have attracted enormous interest with respect to maximizing user experiences in daily life[1]. In particular, flexible displays with a free form factor can be conformally attached to arbitrary curvilinear surfaces of clothes, automobiles, and skins to realize user-friendly visual IoT platforms[2–4]. Among them, inorganic micro light-emitting diodes (μLEDs) have been spotlighted as a promising candidate to replace

conventional light sources for flexible displays and consumer electronics due to their superior thermal/mechanical stabilities, and outstanding electrical/optical properties (e.g., contrast ratio, brightness, response time, and power efficiency)[5–9]. Recently, μLED devices such as ultrahigh-resolution televisions, augmented/virtual reality (AR/VR) glasses, and epidermal patches were introduced at the Consumer Electronics Show (CES). However, these state-of-the-art products still

[1]Department of Materials Science and Engineering, Korea Advanced Institute of Science and Technology (KAIST), 291 Daehak-ro, Yuseong-gu, Daejeon 34141, Republic of Korea. [2]Division of Advanced Materials Engineering, Jeonbuk National University, 567 Baekje-daero, Deokjin-gu, Jeonju-si, Jeollabuk-do 54896, Republic of Korea. [3]Department of Mechanical Engineering, Korea Advanced Institute of Science and Technology (KAIST), 291 Daehak-ro, Yuseong-gu, Daejeon 34141, Republic of Korea. [4]BSP Co., Ltd., 41-4, 170 Burim-ro, Dongan-gu, Anyang-si, Gyeonggi-do 14055, Republic of Korea. [5]These authors contributed equally: Sang Hyun Park, Tae Jin Kim, Han Eol Lee. ✉e-mail: keonlee@kaist.ac.kr

have a significant problem of high prices impeding mass-commercialization, owing to the error-prone processes, complex manufacturing systems, and low production yield[5,6,10,11]. In addition, because of the thermal and chemical vulnerability of polymer materials, flexible μLEDs cannot be directly fabricated onto plastic substrates by conventional high-temperature processes. To overcome these intrinsic limitations, micro-sized inorganic semiconductors have to be transfer-printed from the rigid mother wafers to the targeted universal substrates[12–14].

Transfer technologies have been developed to rearrange the μLEDs grown on a wafer scale into large display panels with a sparse distribution. For the commercialization of μLED displays, mass transfer with high selectivity and controllability is essential for multiple transfer printing of massive RGB μLED dies in desired layouts to reduce the final product costs[6,10,15–20]. Several approaches including elastomeric transfer printing, laser-assisted transfer, electrostatic/electromagnetic transfer, and fluidic self-assembly methods have been developed to assemble microchips onto the target substrates. Elastomeric transfer printing utilizes an elastomeric stamp (Polydimethylsiloxane, PDMS) to pick up and release the semiconductor inks from the donor wafer to the target substrate. The main transfer mechanism of elastomeric transfer printing is the kinetic control of van der Waals force between the PDMS stamp and microchips, which is modulated by the stamp peeling rate[15,21–28]. Laser-assisted transfer uses a laser beam to generate the local blister (or ablation) on the dynamic release layer, which temporarily holds the μLEDs. These laser-induced ablation and blister induce the μLEDs to release on final substrates from the carrier substrates[14,29,30]. Electrostatic transfer utilizes the electrostatic adhesion force to pick up the μLED dies from the mother substrate. The electrostatically charged transfer head attracts and releases the μLEDs by modulating the applied voltage. Electromagnetic transfer uses electromagnetic attraction force, which is generated by the coil in the transfer head and magnetic layer deposited on the μLEDs[18]. Fluidic self-assembly integrates μLEDs on the target substrate via gravity and capillary forces, which drive and capture the microchips on the binding sites. The μLEDs, which are dispersed in a fluid such as isopropanol, acetone, or water, are captured on the binding sites of the target substrate, followed by the bonding process for electrical interconnection[17–19]. Although there have been successful demonstrations of μLED displays, these transfer methods still suffer from critical issues such as the need for additional adhesives, stamp reusability, chip damage, misalignments, and poor selectivity[20–35]. Physical suction force induced by an air pressure difference is also utilized for the pick-and-place of LEDs[36]. However, selective modulation of the micro-vacuum should be investigated for reliable transfer of a large number of μLEDs with low cost (see Supplementary Table 1 for details).

Micro-hole (μ-hole) drilling on a glass substrate has been widely exploited for the commercial products of packaging interposers, and biochips due to its excellent electrical isolation properties, optical transparency, and mechanical/chemical stabilities[37,38]. Conventional drilling techniques such as mechanical drilling, dry etching, and laser drilling are unsuitable for precise dimension control of μ-holes with a high aspect ratio because of crack formation, limited minimum hole size, and surface roughening[39–41]. To overcome these obstacles, laser-induced etching (LIE), consisting of laser material modification and subsequent wet etching, has been proposed to fabricate complex 3-dimensional (3D) structures in transparent materials including quartz, sapphire, and glass with a high speed of up to 10,000 holes per second. The laser irradiation locally induces structural defects within the focal volume, causing an etch rate difference (5-50 times higher than the unirradiated zone) between the laser-affected zone (LAZ) and the unaffected zone[42–44].

Herein, micro-vacuum assisted selective transfer (μVAST) of inorganic thin-film semiconductors was introduced for realizing high-density soft electronics on unusual substrates. The LIE methodology on glass substrates enabled the production of a large number of μ-hole arrays (20-50 μm size) with a high aspect ratio (>5). The vacuum controllable module (VCM), composed of micro-channels (μ-channel) on top of μ-holes, was constructed by microelectromechanical systems (MEMS) technology to selectively control the micro-vacuum force for the pick-and-place of microchips. By using the VCM, the μVAST achieved selective and massive transfer of thin-film semiconductors with high adhesion switchability of $3.364 \times 10^6$, three orders of magnitude higher compared to other transfer technologies. The transfer mechanism and reliability of the μVAST were theoretically investigated by a finite element method (FEM) simulation to adjust the adhesion force balance during the pick-up and release steps. Various inorganic thin-film semiconductors including silicon and III-V semiconductors were transfer-printed from the donor wafers to final substrates via micro-vacuum suction without any additional adhesives. Heterogeneous integration and selective transfer printing based on μVAST were realized with diverse device shapes, sizes, and thicknesses. Finally, a high-performance flexible μLED device was demonstrated on a polyimide (PI) substrate with an average transfer yield of 98.06%, showing uniform optical power intensity and mechanical stability (<9% performance degradation under harsh bending conditions).

## Results

### Concept of micro-vacuum assisted selective transfer (μVAST)

Figure 1a–c schematically illustrates the overall concept of μVAST to transfer-print microscale semiconductors from donor wafers to a targeted substrate via micro-vacuum force. Figure 1a shows an exploded sectional illustration of a VCM composed of the hard-polydimethylsiloxane (h-PDMS) μ-channel and LIE-drilled glass. The interdigitated μ-channels were engraved on the h-PDMS block through a molding process of soft lithography[45]. The μ-holes, periodically arranged with a regular distance of 600 μm, were located under the μ-channel. Figure 1b exhibits the working principle of the μVAST procedure. The air pressure inside each μ-channel was individually controlled by an external vacuum pump, enabling selective micro-vacuum transfer printing. In the pick-up process, the suction force was generated at each μ-hole to separate microchips from a donor wafer by switching the μ-channel pressure level from the vent state ($P_{\text{μ-hole}} = P_{\text{atm}}$) to the vacuum state ($P_{\text{μ-hole}} < P_{\text{atm}}$). For the releasing procedure, the suction force was immediately removed by venting the μ-channel ($P_{\text{μ-hole}} = P_{\text{atm}}$) to print the devices onto the target substrate. Figure 1c shows that microchip arrays with different sizes and shapes could be selectively lifted from diverse mother wafers by micro-vacuum suction force. Subsequently, the exfoliated chips were released onto the desired substrate with precise alignment, realizing heterogeneous integration of inorganic semiconducting materials on a single substrate. The movie of entire process including LIE and μVAST is shown in Supplementary Movie 1. Si thin-films with three different shapes and sizes including pentagon (red), circle (green), and star (blue) shapes were transfer-printed onto a deformable PI substrate with accurate position, as displayed in the colored scanning electron microscopy (SEM) image in Fig. 1c. Figures 1d, e show SEM images of AlGaInP thin-film μLED arrays before and after the μVAST process, respectively. For retrieval of μLEDs via micro-vacuum force, μLED arrays were fabricated in a freestanding state, anchored to the unetched part of a wafer by a thin micro-bridge (μ-bridge) (see Methods and Supplementary Fig. S3 for fabrication details)[23,26–28]. During the vacuum state ($P_{\text{μ-hole}} < P_{\text{atm}}$), the μLED arrays were separated from the donor wafer and attached to the VCM with the fracture of μ-bridges. These thin-film μLED arrays, picked up by the VCM, were transfer-printed onto an anisotropic conductive film (ACF)-pasted PI substrate.

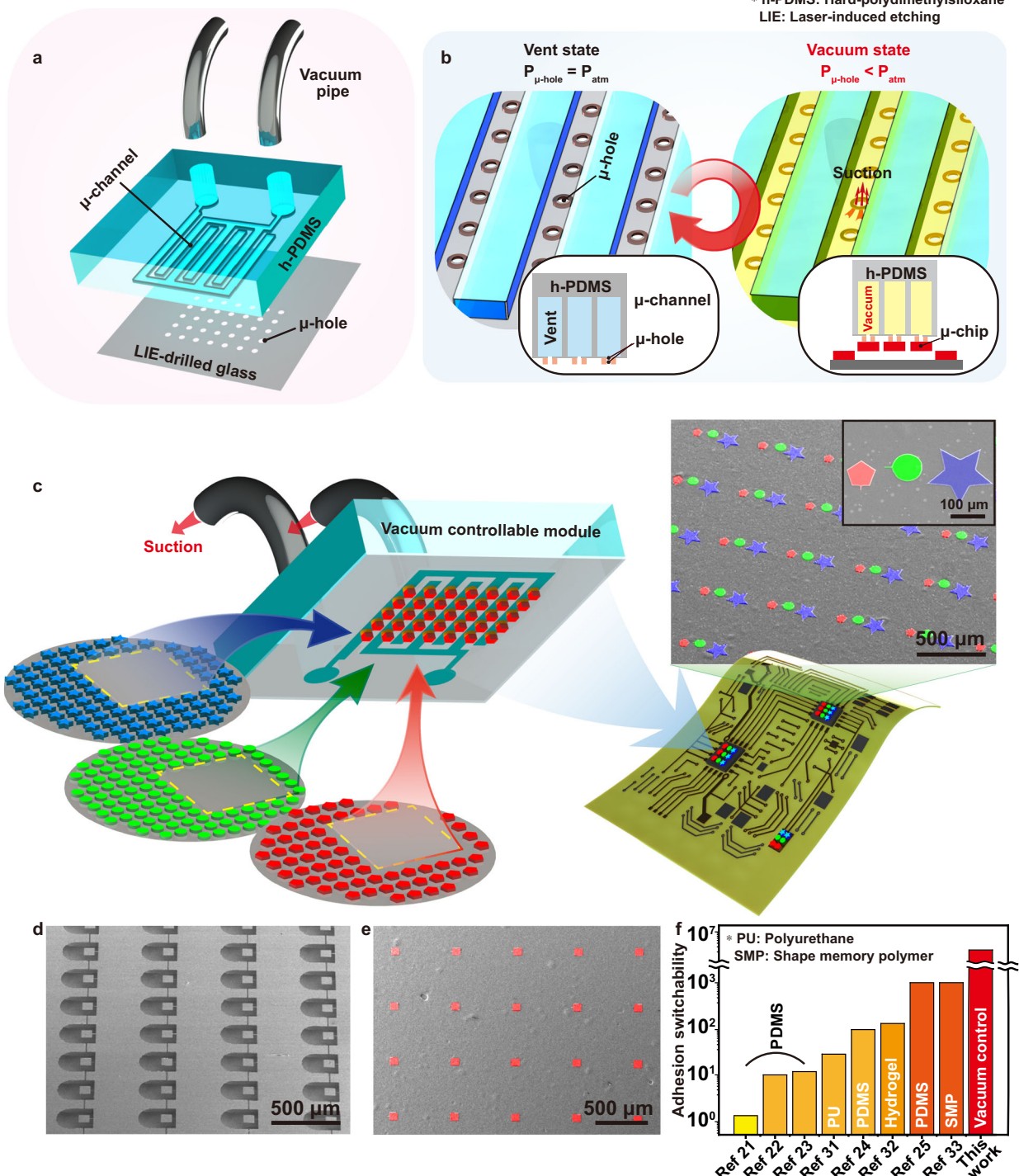

**Fig. 1 | Concept of micro-vacuum assisted selective transfer printing (μVAST). a** An exploded scheme of the VCM with interdigitated μ-channels and μ-holes arranged with regular distances. **b** Pressure change inside the μ-channel during the μVAST process. **c** Schematic of μVAST procedure for heterogeneous integration by selectively picking-up microchips from diverse wafers. Colored SEM image of transferred Si arrays with various shapes including pentagon, circle, and star shapes. **d** SEM image of freestanding μLED arrays on GaAs substrate. **e** Colored SEM image of transferred μLEDs on the final substrate. **f** Comparison of adhesion switchability of μVAST with other transfer printing methods.

Adhesion switchability is an important factor in achieving high transfer yield and controllability in the pick-up and releasing process[20]. Figure 1f exhibits a comparison of adhesion switchability between μVAST and previously reported transfer printing technologies.

In the case of μVAST, the adhesion switchability was the ratio of maximum adhesion force to minimum adhesion force, which was

determined by the following Eq. (1)[20].

$$\text{Adhesion switchability} = \frac{F_{\max}}{F_{\min}} = \frac{F_{\text{suction}} + F_{\text{vdW}}}{F_{\text{vdW}}} \quad (1)$$

where $F_{\max}$ is the maximum adhesion force, which is expressed as the sum of the micro-vacuum suction force and the van der Waals force.

$F_{\text{suction}}$ is the vacuum suction force generated at the μ-holes and $F_{\text{vdW}}$ is the van der Waals force between the microchips and the VCM. On the other hand, $F_{\text{min}}$ is the minimum adhesion force, which is defined by only the van der Waals force without the vacuum suction force, because the vacuum suction force is removed by venting the μ-channel. $F_{\text{vdW}}$, the van der Waals interaction between two contact surfaces, could be calculated by Eq. (2)[46].

$$F_{vdW} = \frac{A}{6\pi D^3} \times \text{Contact area} \qquad (2)$$

where $A$ is the Hamaker constant and $D$ is the separation distance between the microchips and the VCM[46]. Hamaker constant ($A$) is a physical coefficient to define the van der Waals interaction between two contact bodies[47]. Separation distance ($D$) is measured by an atomic force microscopy (AFM) analysis because the separation distance of two contacted surfaces is determined by the roughness of each surface[46]. According to the AFM results (Supplementary Fig. S4), a μLED had a relatively large surface roughness value compared to that of the VCM. In this case, the separation distance was equivalent to the distance between a smooth surface and a rough surface, which was the maximal roughness peak ($d_{\text{max}}$) of a rough surface. Based on this approximation, the separation distance between the μLED and the VCM was expressed as below.

$$D \approx d_{\text{max}} \text{ of μLED surface} \qquad (3)$$

The contact area between the μLED and the VCM (Supplementary Fig. S5) could be calculated as follows:

$$\text{Contact area} = \text{Area of μLED} - \text{Inner circle area of μ}-\text{pillar} \qquad (4)$$

Based on Eq. (2) ~ (4), the $F_{\text{vdW}}$ of μVAST was 85 pN. As the $F_{\text{suction}}$ of 286 μN was generated by an air pressure difference between the inside and outside of the μ-channel during the pick-up process, the adhesion switchability of μVAST could be calculated as $3.364 \times 10^6$ by Eq. (1). The adhesion switchability of μVAST was three orders of magnitude higher compared to that of previously reported transfer methods, thus demonstrating the controllability and reliability of μVAST (see Supplementary Note 1 for calculation details).

## Laser-induced etching (LIE) for high aspect ratio μ-hole arrays

Through-glass fabrication of μ-hole arrays should be accomplished with accurate position and uniform hole size for reliable μVAST. Figure 2a presents a schematic diagram of the LIE procedure to make 20 μm-sized μ-hole arrays with an aspect ratio of 5:1 on a glass substrate. A Femtosecond laser Bessel beam with a wavelength of 1064 nm was irradiated on the glass to induce photo-modification inside the silica materials. The long focal depth of the Bessel beam, which had a uniform intensity distribution along the entire glass thickness, caused the formation of homogeneous morphological defects along the longitudinal direction[48]. When the glass substrate was exposed to a laser Bessel beam with an ultrashort pulse, the laser-induced optical breakdown delivered the photon energy to the lattice by multiphoton and avalanche ionization of a large number of electrons[38,42,43]. Due to this energy transfer, microexplosions with extremely localized heat and pressure occurred inside the glass, inducing permanent morphological changes such as voids and cracks[48,49]. These photo-induced defects accelerated the penetration of chemical etchants during the subsequent wet etching process, causing a high etch selectivity of the LAZ compared to the unaffected zone[43]. This structural LAZ modification enabled the fabrication of 20 μm-sized holes with a high aspect ratio (>5) on the glass substrate. To punch μ-hole arrays with a diameter of 20 μm, a femtosecond infrared (IR) laser was irradiated along the circumference of μ-holes,

forming a porous structure in silica[38]. This porous circumference melted rapidly during the subsequent wet etching process to separate the inner part of the μ-holes from the glass substrate. Figure 2b shows 3D X-ray microscopy (3D XRM) images of the laser-irradiated glass substrate to investigate structural defects inside the LAZ. As shown in the top and side views of the 3D XRM images, laser shots with a beam size of 2.7 μm were periodically focused onto the glass at a regular distance of 3 μm. Laser Bessel beams ablated the silica materials, generating sub-micrometer scale voids on the glass surface, as presented in the inset of the top 3D XRM image. In addition, the voids and cracks inside the LAZ were vertically aligned throughout the entire thickness of the laser-irradiated glass, as displayed in the inset of the side 3D XRM image. Figure 2c exhibits a comparison of the porosity in silica materials between the LAZ and the unaffected zone. The porosity dramatically increased from 0.15 vol% to 3.09 vol%, 20 times higher after the laser modification process, leading to an increase in the diffusion rate of chemical etchants through the LAZ. Figure 2d shows a comparison of the chemical etch rate between the LAZ and the unaffected zone. The chemical etching rate of LAZ was 36 times higher than that of the unaffected zone, which is beneficial for realizing a high aspect ratio (>5) of μ-hole arrays on glass substrates. Figure 2e exhibits a top-view SEM image of μ-hole arrays with 20 μm diameter and 600 μm pitch formed by the LIE method on a glass substrate. The μ-hole arrays completely penetrated the 100 μm thick glass substrate without any dimensional inaccuracy, positional error, or surface damage.

## μVAST mechanism through VCM

Figure 3a displays an optical image of the VCM, consisting of h-PDMS μ-channels, attached to a LIE-drilled glass substrate. For clear identification, two separate μ-channels were filled with red ink, confirming the complete isolation of each μ-channel for independent pressure control. h-PDMS with high shore hardness was used for the VCM in order to prevent the collapse of polymer μ-channel in a depressurized state[45]. The μ-hole arrays were fluidically connected with each other through vacuum channels to simultaneously apply suction force to a large number of microchips, as shown in the inset of Fig. 3a. Figure 3b shows a tilted SEM image of μ-hole arrays, the direct contact interface with microchip arrays during the μVAST process. In order to maximize the adhesion switchability, SU-8 micro-pillars (μ-pillar) were formed around the μ-hole arrays using the standard photolithography method. The μ-pillars enhanced the suction force by reducing the air leakage derived from defects or contamination at the interface. In addition, the μ-pillars enabled minimal contact area between the VCM and microchips to minimize the van der Waals force during the printing process. Figure 3c presents a cross-sectional SEM image of the VCM with μ-channels on a LIE-drilled glass substrate. The μ-holes were aligned under the μ-channels to deliver the micro-vacuum suction force, caused by the air pressure difference between the inside and outside of the μ-channels. The LIE-drilled glass was treated with oxygen plasma to enhance the adhesion force to h-PDMS, minimizing inter-channel interference induced by air leakage (see Methods and Supplementary Figs. S1 and S2 for fabrication details)[50].

The competing mechanisms between the micro-vacuum suction force and fracture force of the μ-bridge are critical for achieving a reliable pick-up procedure from the donor wafer with a high yield[20]. The fracture force of a μ-bridge was experimentally measured by a nano-indentation test, as shown in Fig. 3d. μ-bridges with three different widths of 1.00, 1.25, and 1.50 μm were broken by indentation force of 36.5, 49, and 56 μN, respectively. Figure 3e shows the stress distribution during the pick-up process determined by FEM calculation. When a μLED was lifted up by the micro-vacuum force, the maximum principal stress of 669 MPa was concentrated at the end of the μ-bridge for fracture. Figure 3f presents a comparison of the micro-vacuum suction

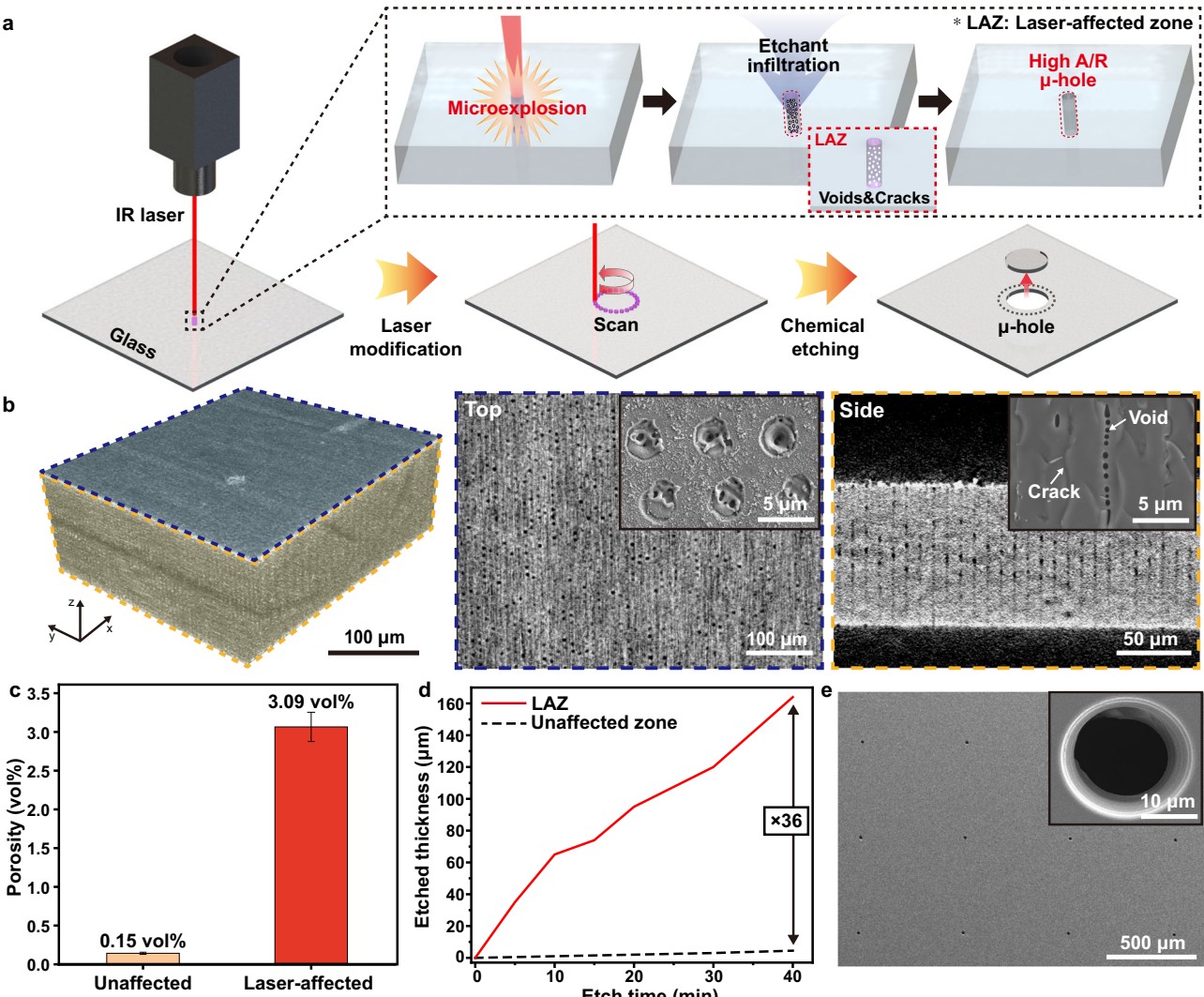

**Fig. 2 | Laser-induced etching (LIE) of μ-hole arrays on a glass substrate.**
**a** Schematic illustration of LIE technology on a glass substrate. **b** 3D XRM image of a laser-irradiated glass. Top and side view XRM image of the laser-irradiated glass. The right upper inset shows an SEM image of laser shots on top of a glass and laser-induced defects inside a glass substrate, respectively. **c** Comparison of porosity between LAZ and unaffected zone. The data point indicates the mean value and the error bars indicate the standard deviation. **d** Comparison of chemical wet etch rate between LAZ and unaffected zone. **e** SEM image of 20 μm-sized μ-hole arrays on a glass substrate. The right upper inset shows a magnified SEM image of a through-glass μ-hole.

force with the critical fracture force of μ-bridges depending on the bridge widths. The suction force was defined by Eq. (5).

$$F_{suction} = (Pa - Pc) \times \text{Suction area} \qquad (5)$$

where $P_a$ and $P_c$ are the outside and inside air pressures of the μ-channel, respectively. Based on Eq. (5), the micro-vacuum suction force was 286 μN because $P_c$ decreased to 316 mTorr in a depressurized state. According to the stress distribution simulation, the fracture forces for retrieval were 157, 184, and 212 μN for bridge widths of 1.00, 1.25, and 1.50 μm, respectively. The calculated fracture force of μ-bridges matched the trend of the nano-indentation results, showing a much lower value than the suction force of 286 μN. The little difference in fracture force between the nano-indentation measurement and pick-up simulation was derived from the different fracture behavior and stress distribution of each case (Supplementary Fig. S6). Nevertheless, these results confirmed that the micro-vacuum suction force was large enough to break the μ-bridge for reliable transfer printing. Figure 3g displays optical microscopy (OM) images of the overall μVAST process, including the alignment, contact, lift-up, and release

procedures. The optically transparent VCM enabled clear observation of the entire μVAST and alignment processes. During the pick-up step, the center of a μ-hole was aligned with the freestanding μLED on a donor wafer through the transparent VCM. After conformal contact between the VCM and the μLED, the μLED was separated from the donor wafer and attached to the VCM by a micro-vacuum force. Finally, the μLED was released onto the target location by venting the μ-channel to remove the suction force (see Supplementary Movie 2 for the pick-up and release procedures of μVAST). The entire μVAST process was performed by customized μVAST equipment, consisting of eight independent vacuum lines, moving stages with XYZ axes, and a tilting system of three axes, to enable high positional accuracy of selective transfer. The three axes for the tilting system were composed of XZ/YZ planes tilting and XY rotation for conformal contact and alignment between the VCM and microchips (Supplementary Fig. S7). Figures 3h, i display the pressure difference and suction force depending on the diameter and number of μ-hole, respectively. The pressure difference and suction force decreased with an increasing diameter and number of μ-hole because of unintentional misalignment. However, the micro-vacuum suction force decreased by 0.78%

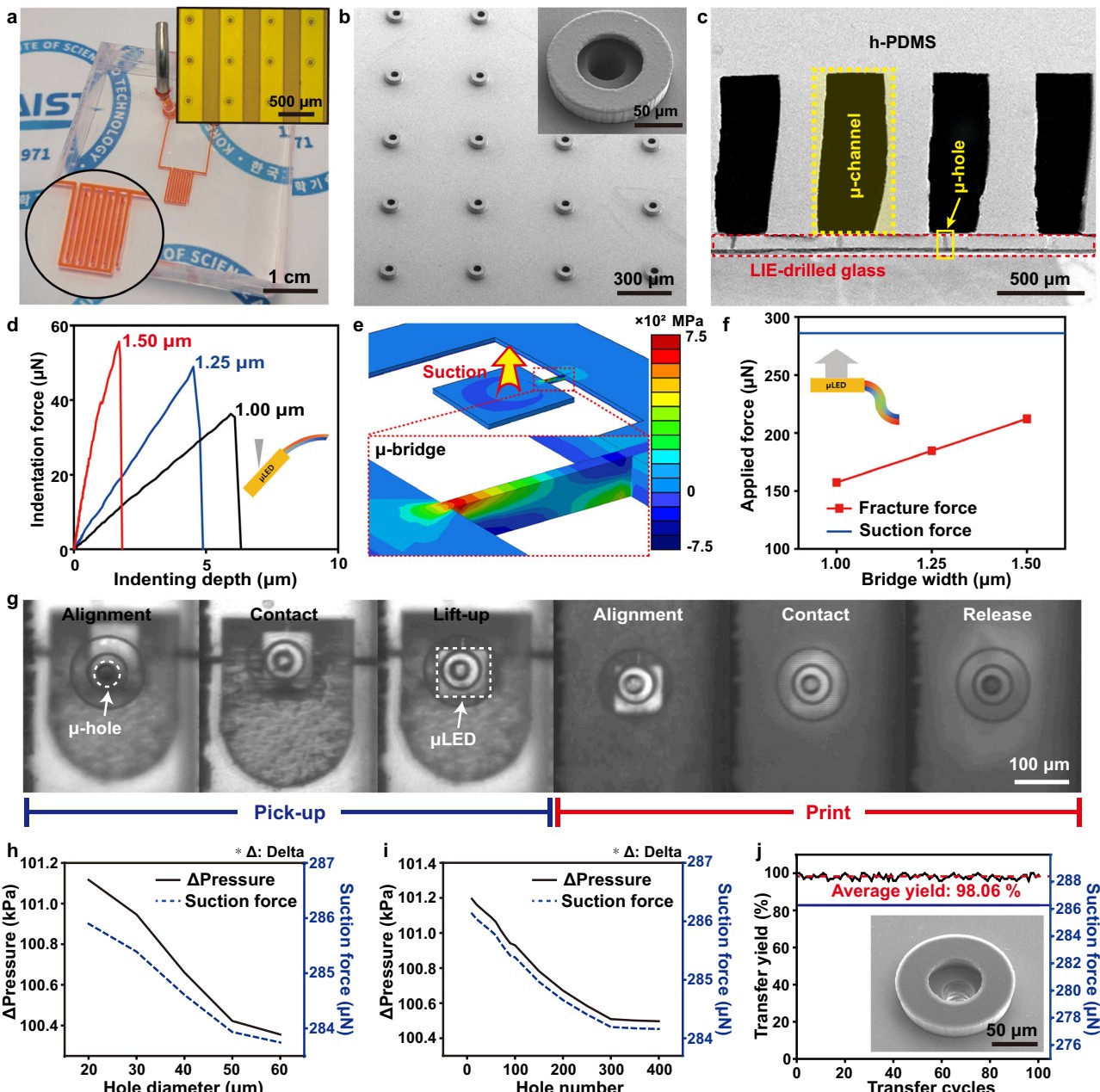

**Fig. 3 | Mechanism analysis of μVAST. a** Optical image of the VCM. The right upper inset shows an OM image of μ-channels and μ-holes. **b** SEM image of μ-holes surrounded by ring-shaped μ-pillars. The inset displays the magnified SEM image of a μ-pillar. **c** Cross-sectional SEM image of VCM with μ-holes aligned at the center of each μ-channel. **d** Nano-indentation results of μ-bridges to experimentally investigate the fracture forces. The inset displays the fracture behavior during the nano-indentation. **e** Calculated stress distribution at the μ-bridge during the pick-up process. **f** Comparison of suction force and fracture force of μ-bridge depending on the bridge width. The inset shows the fracture behavior during the pick-up process. **g** OM images in each step of μVAST. (i) Alignment, (ii) Contact and depressurization, (iii) Lift-up, (iv) Alignment with target substrate, (v) Contact and vent, (vi) Release. **h** The pressure difference and suction force depending on the diameter of μ-hole. **i** The pressure difference and suction force depending on the number of μ-hole. **j** Average transfer yield and suction force during 100 μVAST cycles. The inset shows the SEM image of a μ-pillar after 100 μVAST cycles.

and 0.64% depending on the increase in diameter and number of μ-hole, respectively, showing a much higher value than the fracture force of the μ-bridge. Figure 3j shows the average transfer yield and suction force during 100 μVAST cycles of 10 × 10 μLED arrays. The average transfer yield of μVAST exhibited a high value of 98.06% during the repeated test. Furthermore, the μVAST maintained a constant suction force of 286 μN, enabling repetitive transfer printing with a high yield. This scalability and reliability of μVAST are attributed to the structural robustness of a large number of μ-holes in the LIE-drilled glass substrate.

## Universal transfer of thin-film semiconductors via μVAST

The versatility of μVAST technology was validated by selectively transferring microchips with various materials and dimensions onto diverse substrates. Figure 4a shows μLEDs transfer-printed on a flexible PI substrate with negligible misalignment. The average positional error was 7.5 μm and 4.6 μm in the lateral and vertical directions, respectively (Supplementary Fig. S8). In order to minimize alignment errors during μVAST, the auto-alignment system is being developed to upgrade the customized μVAST equipment. The transfer-printed 10 × 10 μLED arrays were

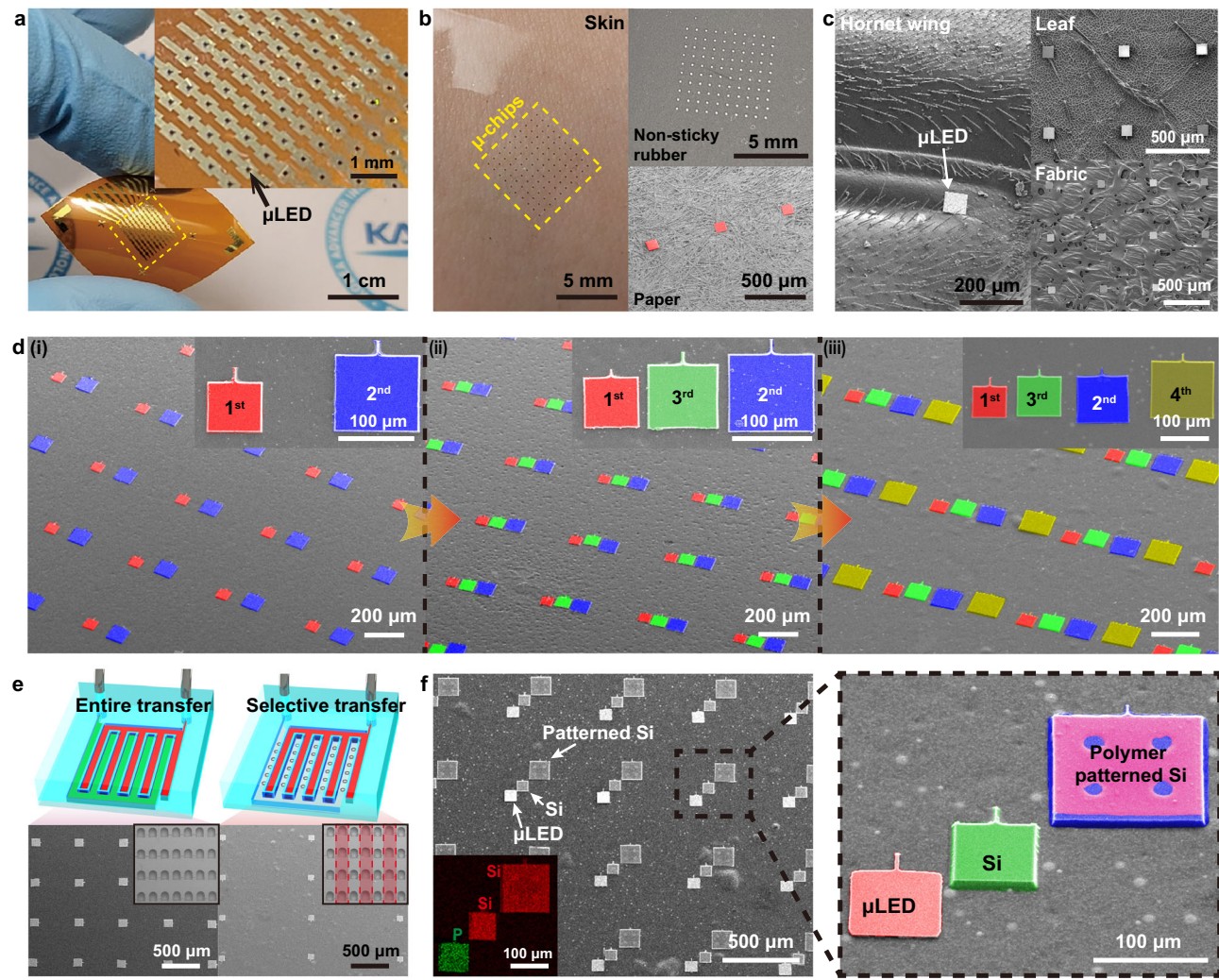

**Fig. 4 | Universal transfer printing of thin-film semiconductors via μVAST.**
**a** Optical image of transferred μLEDs on a flexible PI substrate. **b** Optical and SEM images of transferred μLEDs on various substrates including human skin (left), non-sticky rubber sheet (right upper), and paper (right lower). **c** SEM images of transferred μLEDs on a hornet wing, leaf, and fabric. **d** Colored SEM images of multiple transfer processes with various chip sizes. (i) Transfer printing of $80 \times 80\ \mu m^2$ and $120 \times 120\ \mu m^2$ Si chips with a certain distance for 3rd transfer. (ii) 3rd transfer (Si chips with $100 \times 100\ \mu m^2$ size) between the previously transferred Si chips. (iii) 4th

transfer of $140 \times 140\ \mu m^2$-sized Si chips on the right side of previously transferred chips. **e** Schematics and results of selective transfer printing enabled by independent pressure control of each μ-channel. The inset shows SEM images of a donor substrate after selective transfer. **f** SEM images of heterogeneous integration of various microchips. The inset shows the EDS mapping results of AlGaInP μLED, bare Si, and polymer-patterned Si chips on the same plane. The right image exhibits a magnified colored SEM image of thin-film semiconductors with diverse sizes, materials, and thicknesses, transfer-printed on the same substrate.

electrically interconnected with the bottom electrode through ACF[2,6,8]. In addition, the μLED arrays were successfully transfer-printed onto arbitrary substrates such as human skin, non-sticky rubber, paper, hornet wing, leaf, and even a fabric regardless of the surface adhesion forces and morphologies of the target substrates, as shown in Fig. 4b, c. Furthermore, not only the flat semiconductor microchips, non-planar objects with rough surfaces were transfer-printed through μVAST, as shown in Supplementary Fig. S9. Figure 4d demonstrates the multiple transfer printing of microchip arrays via μVAST. First, $80 \times 80\ \mu m^2$ and $120 \times 120\ \mu m^2$-sized silicon (Si) thin-film arrays were sequentially transfer-printed onto PI film, as shown in Fig. 4di. Subsequently, $100 \times 100\ \mu m^2$-sized Si thin-films (3rd transfer) were released between the 1st and 2nd transferred chips (Fig. 4dii). Finally, $140 \times 140\ \mu m^2$-sized Si arrays (4th transfer) were printed at the right side of the 2nd Si arrays, enabling a horizontal arrangement of four different-sized microchips on the final substrate, as presented in Fig. 4diii. Controllability of transfer arrangement with

diverse chip sizes could be utilized for RGB full-color μLED layout[12,16,29]. Figure 4e presents schematics and SEM images of the selective transfer of μLED arrays, achieved by independent pressure control of the μ-channels. Two interdigitated μ-channels in the VCM, displayed in red and green colors, were simultaneously depressurized for the pick-up process of μLED arrays with a lateral distance of 600 μm, as shown in the left SEM image of Fig. 4e. For selective transfer, only half of the μLED arrays were selectively picked up and released onto the target substrate with a lateral pitch of 1200 μm by controlling the vacuum state of one μ-channel, as shown in the right SEM image of Fig. 4e. Figure 4f demonstrates the heterogeneous integration of thin-film microchips, including AlGaInP, Si, and polymer-patterned Si through μVAST. μLED, Si, and polymer-patterned Si arrays with various sizes and thicknesses were heterogeneously assembled on the same substrate in a diagonal arrangement. As presented in the inset of Fig. 4f, the phosphorus in the AlGaInP layer and silicon elements were detected by energy dispersive spectroscopy (EDS)

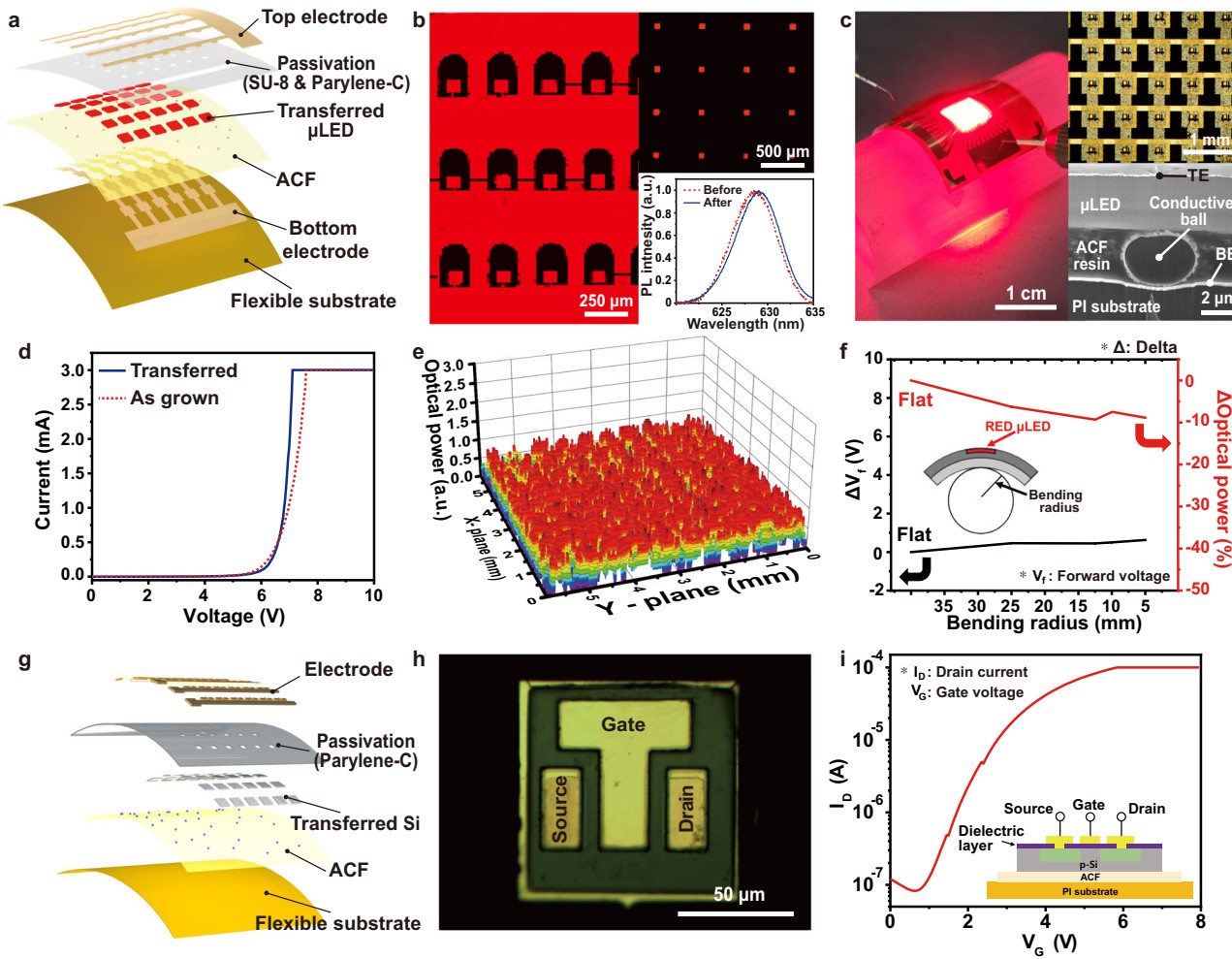

**Fig. 5 | Flexible devices fabricated by µVAST. a** Exploded illustration of vertical thin-film µLEDs on PI substrate. **b** The photoluminescence (PL) maps of µLED arrays before (left) and after (right upper) µVAST. The right lower graph shows the PL spectrum of µLEDs before and after µVAST. **c** A photograph of a flexible µLED attached to a cylinder (bending radius: 1 cm). The right upper shows an OM image of 10 × 10 arrays of 80 × 80 µm²-sized µLED. The right lower exhibits a cross-sectional SEM image of transfer-printed µLED packaged via ACF. **d** I–V characteristics of transferred µLEDs and µLEDs on the as-grown wafer. **e** Luminescence distribution of 10 × 10 µLED arrays transfer-printed on a PI substrate. **f** Forward voltage and normalized optical power change of flexible µLEDs depending on bending radius. **g** Exploded schematic of flexible Si transistors on PI substrate. **h** OM image of a single cell Si transistor, transfer-printed via µVAST. **i** The transfer characteristics ($I_D$–$V_G$) of the flexible transistor.

## Flexible µLEDs & Si transistors fabricated by µVAST

Figure 5a presents a cross-sectional schematic illustration of flexible vertical thin-film µLEDs fabricated by µVAST. Here, 10 × 10 µLEDs were transfer-printed on an ACF-laminated PI film, followed by a thermo-compressive bonding process with 20 kgf pressing and 270 °C heating for vertical electrical interconnection. After the ACF bonding between the µLEDs and bottom electrodes, the transferred µLEDs were passivated by transparent parylene-C polymers, followed by top electrode formation. The left and right upper images of Fig. 5b present photoluminescence (PL) mapping images of thin-film µLED arrays before and after µVAST, respectively. The red color in the PL images confirmed that imperceptible damage occurred in the active AlGaInP layers of the transferred µLEDs during the µVAST process[51]. As shown in the right lower graph of Fig. 5b, there was a negligible change in the PL spectrum of the µLED before and after µVAST. Figure 5c exhibits the 10 × 10 flexible µLEDs, stably emitting bright red light in a bent state with a 1 cm bending radius. The right upper image displays a magnified OM image of AlGaInP µLED arrays while

the right lower image exhibits a cross-sectional SEM image of a transferred µLED, electrically interconnected to the bottom electrodes via ACF. The vertical electrical path was formed by the conductive Au/Ni particles inside the thermo-compressed ACF. Figure 5d compares the I-V characteristics of the µLEDs before and after the µVAST. The forward voltage of the transferred µLEDs was 7.3 V, showing an imperceptible change of 0.3 V compared to that of the µLEDs on the as-grown wafer. This variation in forward voltage is attributed to the change of the contact resistance between the n-type GaInP layer and bottom electrodes[28]. The light distribution of the 10 × 10 transferred µLEDs was investigated by 2D color mapping, as shown in Fig. 5e[9]. The µLEDs exhibited uniform optical power intensity in the overall area of 7 × 7 mm², confirming low variation in the optical properties among the entire transferred µLEDs. The mechanical stability of the flexible µLEDs was investigated through a mechanical bending test under various bending radii, as presented in Fig. 5f. Despite severe bending conditions of up to a 5 mm bending radius on the curved surface, the optical power only decreased by 9.3%, and the forward voltage increased by 0.63 V. Furthermore, as shown in Supplementary Fig. S10, the optical power and forward voltage of the flexible µLEDs changed within 9% and 0.41 V during 10⁴

bending cycles, respectively, demonstrating the mechanical stability of the µVAST-based flexible µLEDs. Figure 5g presents a layer-by-layer schematic illustration of flexible Si transistor arrays on a PI film. The single crystalline Si thin-film arrays on a silicon-on-glass (SOG) wafer were transferred onto a flexible substrate by removing the underlying silicon dioxide ($SiO_2$) layer. Prior to µVAST of free-standing Si arrays, the source/drain regions were doped with phosphor by spin-on-dopant (SOD) and activated by XeCl excimer laser irradiation[52]. A hafnium dioxide ($HfO_2$) dielectric layer was then deposited, followed by Cr/Au formation for the source/drain and gate metal. Finally, Si transistors were demonstrated on a PI substrate through µVAST, as shown in Fig. 5h. Figure 5i presents the transfer characteristics ($I_D$−$V_G$) of the flexible Si transistor array, stably operating with a high current on/off ratio of $10^3$. These results demonstrate that µVAST could be utilized to realize hetero-integrated devices for ultrahigh-speed and low power consumption by assembling various semiconductor materials, including III-V compound semiconductors, and Si[53].

## Discussion

In summary, µVAST technology was developed to transfer-print inorganic semiconductor arrays onto unusual substrates by controlling the micro-vacuum suction force. To fabricate µ-hole arrays for micro-vacuum force, LIE technology induced a 36 times higher etching rate of the LAZ than that of the unaffected zone, making 20 µm-sized µ-hole arrays with an aspect ratio of 5:1 on a glass substrate. A VCM, composed of h-PDMS µ-channels and LIE-drilled glass, modulated the adhesion force from 85 pN to 286 µN by vacuum control. The adhesion switchability of $3.364 \times 10^6$ enabled reliable transfer printing with a high transfer yield of 98.06%. The micro-vacuum suction force of 286 µN was large enough to break a µ-bridge during the pick-up process, which was confirmed by a FEM simulation and a nano-indentation test. Microchip arrays with diverse materials and dimensions were transfer-printed onto unconventional substrates such as human skin, rubber, paper, and a hornet wing by customized µVAST alignment equipment. Selective transfer printing was achieved by independently controlling the vacuum state of each interdigitated µ-channel inside the VCM. Heterogeneous integration was accomplished by transfer printing of AlGaInP µLED, bare Si, and polymer-patterned Si on a single substrate. Finally, flexible µLEDs and Si transistors were demonstrated on PI substrates by µVAST, which can be utilized for flexible µLED displays, biopatches, and hetero-integrated devices. Currently, we are developing the LIE-based VCM with an ultrahigh aspect ratio (>15) µ-holes on a quartz substrate for the large-area mass transfer printing of microchips smaller than 20 µm. Furthermore, the µVAST of RGB full-color µLEDs with a flip-chip structure is being investigated for the commercialization of flexible µLED applications. The µVAST can provide essential breakthroughs for realizing a wide range of high-performance inorganic soft electronics.

## Methods

### VCM fabrication

A replica mold with µ-channel structure was fabricated by a conventional photolithography process using SU-8 100 negative photoresist. SU-8 100 was spin-coated on a glass substrate with a spin speed of 500 rpm for 30 s, followed by a baking process at 100 °C for 3 h. After UV exposure and post-exposure baking at 65 °C for 10 min and 100 °C for 1 h, the SU-8 100 layer was developed for 2 h 30 min to define the µ-channel structure. Thereafter, a metal pipe was attached to the end of the µ-channel to fabricate paths that connect the VCM and external vacuum pump. The replica mold was sealed by PDMS (Sylgard 184, Dow Corning) wall, followed by pouring of the h-PDMS mixture (mass ratio of monomer to cross-linker is 1:1) inside the wall. The h-PDMS block with µ-channel structure was detached from the replica mold after curing at 75 °C for 24 h. The µ-pillars on LIE-drilled glass were

fabricated by a photolithography procedure using SU-8 3025 photo-resist. The photolithography process involved spin-coating of SU-8 3025 at 3000 rpm for 30 s and baking at 65 °C for 3 min and 95 °C for 10 min, followed by UV exposure and 75 s development. Finally, the LIE-drilled glass with µ-pillars was attached to the h-PDMS block with the µ-channel structure by utilizing a mask aligner (MDA-8000B, Midas).

### Freestanding µLED and transistor array fabrication

The active AlGaInP epitaxial layers with distributed Bragg reflector (DBR) and GaInP etch stop layers were grown on a GaAs substrate by metal-organic chemical vapor deposition (MOCVD). The Cr (15 nm) and Au (15 nm) layers were deposited as p-ohmic contact on the LED using radio frequency (RF) sputtering, followed by Ni (500 nm) deposition for a dry etching mask. Afterward, the µLED arrays with the µ-bridge structure were defined by inductively coupled plasma reactive ion etching (ICP-RIE) in an environment of $Cl_2$ (60 sccm)/Ar (12 sccm). The plasma was generated by bias of 400 W/100 W and maintained for 150 s at 10 mTorr process pressure. After dry etching, the GaAs layer, which supported the µLED chips, was selectively removed by hydrogen peroxide ($H_2O_2$) and citric acid ($C_6H_8O_7$)-based etchant (volume ratio of 1:1) at a temperature of 50 °C for 1 h. A p-type silicon-on-glass (SOG) wafer (top Si thickness: 7 µm) was used for freestanding transistor arrays. A 300 nm thick $SiO_2$ layer was deposited on the SOG wafer by plasma-enhanced chemical vapor deposition (PECVD) and patterned for source and drain regions by dry etching under a gas condition of $O_2$ (10 sccm)/$C_4F_8$ (25 sccm) for 5 min. Afterward, heavily n-doped regions were formed by spin-on-dopant (P509, Filmtronic) for the source and drain at 3000 rpm spin speed with annealing at 320 °C for 30 min. An XeCl excimer laser (wavelength of 308 nm) was irradiated for dopant activation by double scanning with an energy density of 889 mJ/mm². Si arrays with the µ-bridge structure were then etched by ICP-RIE using $SF_6$ (25 sccm) gas for 18 min, followed by $SiO_2$ undercut using buffered oxide etchant (BOE) (10:1 diluted HF). Finally, $HfO_2$ (15 nm) was deposited for a dielectric layer in the transistor by plasma-enhanced atomic layer deposition (PEALD), followed by electrode metal deposition (Cr: 10 nm/ Au: 50 nm).

### µ-hole arrays fabrication

The borosilicate glass with a thickness of 100 µm (BOROFLOAT® 33, SCHOTT) was used for the fabrication of µ-hole arrays. Laser modification was performed by the femtosecond laser (Monaco 1035, Coherent) Bessel beams with a pulse width of 350 fs and pulse energy of 40 µJ at 1 MHz frequency. After laser irradiation, the laser-irradiated glass was etched by chemical etchant (a mixture of 2.5 wt% HF and 10 wt% $H_2SO_4$) with ultrasonication treatment.

### LIE-drilled glass characterization

The 3D morphological images of laser-irradiated glass were captured by a high-resolution 3D X-ray tomography microscope system (Xradia 620 Versa, Carl Zeiss) with a source voltage of 60 kV.

### Mechanical simulation of µ-bridge fracture

The two-step mechanical simulation was conducted using a commercial finite element method tool (ABAQUS v 6.24). In the first step, the maximum stress at the fracture of the µ-bridge was calculated by simulating the nano-indentation process. In the second step, the required pressure for a reliable pick-up process was evaluated by analyzing the applied stress at the µ-bridge due to the micro-vacuum suction. For the simulation, the unit structure was modeled as a deformable 3D solid with 30,000 nodes and 18,000 elements of type C3D8R (8-node linear brick, reduced integration, hourglass control). The unit structure was considered to be an elastic material (E: 103 GPa, v· 0.31), while its boundary was fixed in all directions.

**Nano-indentation test**

The fracture force of a µ-bridge was measured by a nano-indentation test, using an ultrahigh-resolution nano-indenter with real force and displacement sensors (UNHT³, Anton Paar).

**Device characterization**

The I-V characteristics of flexible µLEDs and transistors were measured by a Keithley 4200-SCS semiconductor parameter analyzer with voltage sweep mode. The electroluminescence spectrum of the µLEDs was measured by AvaSpec-UlS2048-RS optical spectroscopy (Avantes Corp.). The uniformity of optical power was investigated by a 2D color analysis system (CA-2500A) under an injection current of 10 mA. The PL mapping images were captured by a high-resolution PL system (LabRAM HR Evolution Visible_NIR, HORIBA). The cross-sectional image of the device was obtained by means of FIB-SEM (Helios Nanolab 450 F1, FEI) and SEM (SU5000, Hitachi).

**Bending test**. The forward voltage and optical power changes of flexible µLEDs were evaluated on a PDMS mold with various bending radii. The periodic bending test was performed by a customized bending machine (QS48, TPC motion Corp.) for 10,000 bending cycles.

## Data availability

The data that support the findings of this study are present in the article and Supplementary Information. Additional data related to this study are available from the corresponding author upon request.

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

## Acknowledgements

This work was partly supported by the Technology Development Program of MSS (S3207363). This work was also supported by the Wearable Platform Materials Technology Center (WMC) (NRF-2022R1A5A6000846), Convergent Technology R&D Program for Human Augmentation (NRF-2020M3C1B8081519) and RS-2023-00273231 through the National Research Foundation of Korea (NRF) funded by the Ministry of Science and ICT.

## Author contributions

S.H.P., T.J.K., H.E.L., and K.J.L. conceived the idea of µVAST and designed the experiments. S.H.P. and T.J.K. performed the overall experiments and data analysis. B.S.M., M.S., and T.S.K. conducted the numerical calculation. M.S.K., J.H.S., S.H.L., J.H.L., Y.B.K., K.Y.N., and H.J.P. assisted with device fabrication. S.H.P. wrote the manuscript. T.J.K., H.E.L., and K.J.L. helped revise the manuscript. K.J.L. supervised the research and contributed to the discussion of the results.

## Competing interests

The authors declare no competing interests.
