## [Peer Review File · Nature Communications]

REVIEWER COMMENTS

Reviewer #1 (Remarks to the Author):

The authors presented a study regarding a technique to assemble micro-sized semiconductors. The results are clearly explained and shown. This technique shows to be efficient and is interesting. Despite this it would be great if the following is clarified:

- In the introduction the authors mentioned few other techniques available for doing transfer printing, however the techniques are not explained. Proper explanations and comprehensive review, as well as proper referencing shall be added
- Physical suction has been used for pick and place of LEDs. The author mentioned that in the text. Can you please talk more about the originality of the technique used compared to existing methods?
- The authors mentioned that this technique is advantageous regarding its capability to do high volume production and mass commercialization (which is limitation of current techniques) and at affordable price. However, the method followed involves multiple processing steps (during and prior to printing). Therefore a rough cost calculation and comparison with the current techniques would be good to add
- Sometimes subjective words are added in the text like “outstanding” which is better to be avoided and replaced with scientific data. Also sometimes the word “we” is used, whereby the passive voice would be advised.
- It would be good to clearly add the limitations of the current technique (from materials and geometrical perspectives). For example what are the smallest structures that can be manufactured and transferred and what is the percentage error in alignment?)
- Certain bridge widths were chosen and studied, what is the max bridge width that can be used provided the maximum suction force of the micro pump ?
- Regarding controlling the suction force, can you please provide more details about that and the range of forces that can be applied?

Reviewer #2 (Remarks to the Author):

In this manuscript, the authors introduced a micro-vacuum assisted selective transfer (μ VAST) approach for releasing and transferring micro-sized inorganic thin-film semiconductor devices such as micro-LEDs, transistors, etc. The topic is worth investigating and the authors have reported some important results. Specifically, due to the high adhesion switchability of the presented method, the author successfully

transferred some embodiments. This manuscript can be re-evaluated for publication provided the authors sufficiently address the following minor comments.

Comments and suggestions

(1) This approach only presented the release/transfer of planar or relatively flat devices with smooth surfaces. In my opinion, some non-planar geometries of the semiconductor chips will lead to the inapplicability of the proposed transfer printing process. Can authors provide examples of transferring objects which have different roughness, 3D structures, or curved surfaces to demonstrate that this approach is universal?

(2) Because the fracture force of μ -bridge measured/simulated by the authors highly varied with the width change ranging from 1 to 1.5 μm , the geometrical parameters of μ -bridge seem very critical to successfully transferring the semiconductor devices. However, the design or number of these anchors or bridges may vary depending on the lateral dimensions of semiconductor devices. Could the authors provide the required minimum vacuum forces according to more broad width range of μ -bridge (e.g., 1 to less than 30 μm with a certain fixed thickness) without any collapse of the h-PDMS μ -channel?

(3) The fracture forces shown in Figure 3d and 3f do not agree with each other.

(4) Line 223: the authors stated, “the maximum principal stress of 669 MPa was concentrated at the end of the μ -bridge for fracture.” However, comparing with Figure 1c inset, and Figure 4d,e, the fracture locations are not at the junction. How would the authors explain this observation based on their finite element simulation?

(5) The adhesion switchability of this presented approach is quite high. Can the authors mention the range of the micro-vacuum suction force and how they calculated the value in the main text, instead of just in the supplementary note?

(6) Figure 3c: can the authors transfer print an array of μ LEDs onto an uneven surface such as the hornet wing using μ VAST?

(7) Figure 5c: are the μ LEDs individually controllable?

(8) The Outlook section seems more like the conclusion. Please include more discussions on the outlook, challenges and opportunities.

Reviewer #3 (Remarks to the Author):

The paper reported a design of active stamp, which consists of PDMS micro-channels, glass micro-hole arrays, and SU-8 sealing rings around the micro-holes, to control the pick-up and release of microchips

through air pressure. It is shown that the stamp offers a large adhesion switchability, which enables the successful transfer printing of microchips with diverse materials and dimensions onto various unconventional substrates such as human skin, rubber, paper, etc. The work is solid and represents a good advance in the field of transfer printing. Several major issues should be addressed before its consideration for publication.

1. Universal in the title is not appropriate since the proposed approach is only good for flat objects.
2. The reported adhesion switchability is not convincing. The definition of adhesion switchability should be the ratio of the maximum adhesion strength (or force) to the minimum adhesion strength (or force). The suction force could be positive or negative depending on the air pressure, which may yield an infinite adhesion switchability. The reported adhesion switchability in the manuscript is not appropriate and the related part should be revised.
3. What are the factors related to the minimum pitch of μ -hole arrays? Is it possible to transfer devices with small pitch?
4. Can the size of the transferred object be further reduced? What is the limited size of the transferred object?
5. How many μ LEDs are involved in a single test in the transfer yield repeatability test?
6. It is hard to form tightly conformal contact between the hard SU-8 sealing rings and chips. It is better to study whether the sealing ring exists or not on the sealing effect.

[Response to the reviewers]

Reviewer #1:

Overall Comment: *The authors presented a study regarding a technique to assemble micro-sized semiconductors. The results are clearly explained and shown. This technique shows to be efficient and interesting. Despite this, it would be great if the following is clarified:*

Our response: We sincerely appreciate the reviewer for the high evaluation of our manuscript.

Comment 1: *In the introduction, the authors mentioned a few other techniques available for doing transfer printing, however, the techniques are not explained. Proper explanations and comprehensive review, as well as proper referencing, shall be added.*

Our response: We thank the reviewer's helpful comment regarding the additional information on previous transfer printing technologies. Accepting the reviewer's comment, we modified the manuscript to include a more comprehensive introduction to other transfer techniques (e.g. elastomeric transfer, laser-assisted transfer, electrostatic/electromagnetic transfer, and fluidic self-assembly), revised the comparison table in Supplementary Information, and added additional references.

Modification to the manuscript:

- (1) We modified the manuscript to provide additional explanations about previous transfer printing technologies.

“Several approaches including elastomeric transfer printing, laser-assisted transfer, electrostatic/electromagnetic transfer, and fluidic self-assembly method have been developed to assemble microchips onto the target substrates. Elastomeric transfer printing utilizes an elastomeric stamp (Polydimethylsiloxane, PDMS) to pick-up and release the semiconductor inks from the donor wafer to the target substrate. The main transfer mechanism of elastomeric transfer printing is the kinetic control of van der Waals force between the PDMS stamp and microchips, which is modulated by the stamp peeling rate^{15,21-23,25,27,29-31}. Laser-assisted transfer uses a laser beam to generate the local blister (or ablation) on the dynamic release layer, which temporarily holds the μ LEDs. These laser-induced ablation and blister induce the μ LEDs to release on final substrates from the carrier substrates^{14,32,34}. Electrostatic transfer utilizes the electrostatic adhesion force to pick-up the μ LED dies from the mother substrate. The electrostatically charged transfer head attracts and releases the μ LEDs by modulating the applied voltage. Electromagnetic transfer uses electromagnetic attraction force, which is generated by the coil in the transfer head and magnetic layer deposited on the μ LEDs¹⁸. Fluidic self-assembly integrates μ LEDs on the target substrate via gravity and capillary forces, which drive and capture the microchips on the binding sites. The μ LEDs, which are dispersed in a fluid such as isopropanol, acetone, or water, are captured on the binding sites of the target substrate, followed by the bonding process for electrical interconnection¹⁷⁻¹⁹. Although there have been successful demonstrations of μ LED displays,

these transfer methods still suffer from critical issues such as the need for additional adhesives, stamp reusability, chip damage, misalignments, and poor selectivity^{20–35}.” on pages 3-4 of revised manuscript.

(2) We revised Supplementary Table 1 for comparing μ VAST to other transfer technologies.

Supplementary Table 1 | Comparison between μ VAST and conventional transfer printing technologies.

Transfer Method	Principle of Adhesion Control	Adhesion Switchability	Repeatability	Selectivity	Chip Damage	Large-area	Cost
Elastomeric transfer	Kinetic control (Peeling rate) of vdW force	<1000	Moderate	Moderate	Low	Moderate	Low
Electrostatic transfer	Electrostatic force	∞	Good	Moderate	High	Poor	High
Electromagnetic transfer	Electromagnetic force	∞	Good	Good	High	Poor	High
Laser-assisted transfer	Laser heating & Ablation	∞	Poor	Good	High	Good	High
Fluidic self-assembly	Gravity & Capillary force	-	Poor	Poor	Low	Good	Moderate
Pick-and-place	Vacuum suction force		Good	Good	Low	Poor	Moderate
μ VAST	Micro-vacuum suction force	3.364×10^6	High	High	Low	Good	Low

(3) We added additional references about previously reported transfer printing techniques.

18. Chang, W., Kim, J., Kim, M. *et al.* Concurrent self-assembly of RGB microLEDs for next-generation displays. *Nature* **617**, 287–291 (2023).

19. Lee, D., Cho, S., Park, C. *et al.* Fluidic self-assembly for MicroLED displays by controlled viscosity. *Nature* **619**, 755–760 (2023). on page 24 of revised manuscript.

Comment 2: *Physical suction has been used for pick and place of LEDs. The author mentioned that in the text. Can you please talk more about the originality of the technique used compared to existing methods?*

Our response: We thank the reviewer’s helpful comment regarding the originality of micro-vacuum based transfer printing. The conventional pick-and-place method, which utilized vacuum suction force, transferred LED dies one by one using only one vacuum nozzle with a few hundred-micrometer sizes. On the other hand, the μ VAST transferred a large number of μ LEDs in one transfer

cycle by employing lots of micro-sized hole arrays, on which fluidic μ -channels were interconnected for a vacuum suction path. Therefore, we would like to emphasize two important originalities of μ VAST to realize the mass transfer of μ LEDs: 1. Fabrication of μ -hole arrays on a transparent glass/quartz substrate. 2. Selective control of micro-vacuum suction force for selective μ LEDs transfer.

1. Fabrication of μ -hole arrays on a transparent glass substrate

In order to transfer sub-100 μm microchips via micro-vacuum suction force, the 20 μm -sized μ -hole arrays of high aspect ratio (>5) should be fabricated on a transparent glass substrate. This is challenging with conventional drilling techniques such as mechanical drilling, dry etching, and laser drilling due to crack formation on glass substrates, limited minimum hole size, and surface roughening. Instead, we utilized the laser-induced etching (LIE) process to punch the μ -hole arrays of high aspect ratio without any crack, which can be processed with extremely fast speed (10,000 holes per second using femtosecond 1064 nm IR laser). Unlike direct laser drilling, the laser irradiation formed homogeneous morphological defects inside the glass materials, inducing an increase in the chemical etch rate of the laser-affected zone (LAZ). This fast chemical etch rate of LAZ enabled the formation of micro-sized hole arrays without damaging the glass substrate. In addition, the LIE-drilled glass maintains high transparency and a smooth surface, which are critical for the device alignment and conformal contact during the transfer process, respectively.

2. Selective control of micro-vacuum suction force for selective μ LEDs transfer.

For the selective transfer of μ LEDs, μ -hole arrays should be separated into several discrete multi-channels, which are molded on the h-PDMS. The separated μ -channels enabled independent micro-vacuum control of each μ -channel. By utilizing this multi-channel structure, a large number of μ LEDs can be selectively picked-up via independent vacuum modulation of targeted μ -channels.

Adopting the reviewer's comment, we modified Supplementary Information to highlight the originality of μ VAST compared to conventional transfer printing technologies.

Modification to the manuscript:

We revised Supplementary Table 1 to highlight the originality of μ VAST compared to previously reported transfer technologies.

Supplementary Table 1 | Comparison between μ VAST and conventional transfer printing technologies.

Transfer Method	Principle of Adhesion Control	Adhesion Switchability	Repeatability	Selectivity	Chip Damage	Large-area	Cost
Elastomeric transfer	Kinetic control (Peeling rate) of vdW force	<1000	Moderate	Moderate	Low	Moderate	Low
Electrostatic transfer	Electrostatic force	∞	Good	Moderate	High	Poor	High
Electromagnetic transfer	Electromagnetic force	∞	Good	Good	High	Poor	High
Laser-assisted transfer	Laser heating & Ablation	∞	Poor	Good	High	Good	High
Fluidic self-assembly	Gravity & Capillary force	-	Poor	Poor	Low	Good	Moderate
Pick-and-place	Vacuum suction force		Good	Good	Low	Poor	Moderate
μ VAST	Micro-vacuum suction force	3.364×10^6	High	High	Low	Good	Low

Comment 3: *The authors mentioned that this technique is advantageous regarding its capability to do high volume production and mass commercialization (which is limitation of current techniques) and at affordable price. However, the method followed involves multiple processing steps (during and prior to printing). Therefore a rough cost calculation and comparison with the current techniques would be good to add*

Our response: We thank the reviewer’s valuable comment regarding the mass production capability of μ VAST. As the reviewer suggested, we provided a rough cost calculation for mass production of $4 \times 4 \text{ cm}^2$ -sized surface-lighting μ LED patches (Lee et al., *Adv. Healthc. Mater.* **12**, 2201796, 2023) consisting of $100 \times 100 \mu\text{LED}$ arrays using the μ VAST process and compared it to other transfer printing techniques. Transfer printing of $100 \times 100 \mu\text{LED}$ arrays was carried out by four times selective transfer of $50 \times 50 \mu\text{LED}$ arrays onto the $2 \times 2 \text{ cm}^2$ -sized flexible substrates. We assumed that the costs of pre-transfer processes such as wafer manufacturing, μLED chip fabrication, and post-transfer procedures (packaging, and passivation) were the same for all transfer technologies. For this reason, we focused on the calculation of the rough cost for the μ VAST process (μLED chips, equipment, maintenance, and process time) and compared it with other transfer technologies. In addition, we consulted with a μLED manufacturing company in South Korea. However, our preliminary calculations are based on numerous assumptions that may not reflect actual mass production conditions.

As shown in Supplementary Table 2, the calculated production cost per month included prices for the μLED chips, depreciation of equipment, maintenance, and material costs. The monthly

production cost was closely related to the transfer processing speed: the faster the speed, the higher the production volume, and the lower the cost. The equipment depreciation for transfer and other equipment was calculated by dividing the equipment prices by the expected equipment lifetime of 60 months. Maintenance and material costs contained the prices for factory space, labor, and raw materials, depending on product output. We assumed that the elastomeric transfer cannot conduct selective transfer for mass production, and that the equipment costs for laser and electrostatic transfer are higher than those for other transfer technologies.

According to a rough cost calculation, the cost of μ VAST exhibited the lowest level compared to that of other transfer printing technologies. Therefore, this result demonstrates the applicability of μ VAST to μ LEDs mass commercialization. Adopting the reviewer’s comment, we modified Supplementary Information to present the capability of μ VAST in mass commercialization.

Modification to manuscript

We added Supplementary Table 2 to compare the transfer cost between μ VAST and other transfer printing technologies.

Supplementary Table 2 | Comparison of transfer cost between μ VAST and conventional transfer printing technologies.

Transfer process		μ VAST	Pick-and-place	Elastomeric Transfer	Laser-assisted transfer	Electrostatic transfer	
Transfer type		Mass transfer w/ selectivity	Individual transfer	Mass transfer w/o selectivity	Mass transfer w/ selectivity	Mass transfer w/ selectivity	
Monthly production (ea/month)		10,000	1,000	4,000	10,000	10,000	
Production cost (\$/month)	μ LED chips (\$0.004/unit)	400,000	40,000	160,000	400,000	400,000	
	Equipment depreciation (Life: 60 months)	Transfer equipment	4,000	4,000	4,000	8,000	8,000
		Other equipment	11,000	9,000	9,000	11,000	11,000
	Maintenance + Material costs (\$/month)	50,000	37,500	44,000	50,000	50,000	
Cost per unit (\$/unit)		46.5	90.5	54.25	46.9	46.9	

A rough cost for mass production of $4 \times 4 \text{ cm}^2$ -sized surface-lighting μ LED patches (Lee et al., *Adv. Healthc. Mater.* **12**, 2201796, 2023) consisting of $100 \times 100 \mu$ LED arrays using the μ VAST process is calculated and compared to that of other transfer printing techniques. Transfer printing of $100 \times 100 \mu$ LED arrays is carried out by four times selective transfer of $50 \times 50 \mu$ LED arrays onto the $2 \times 2 \text{ cm}^2$ -sized flexible substrates. The costs of pre-transfer processes such as wafer manufacturing, μ LED chip fabrication, and post-transfer procedures (packaging, and passivation) are assumed the same

for all transfer technologies. The calculated production cost per month includes prices for the μ LED chips, depreciation of equipment, maintenance, and material costs. The monthly production cost is closely related to the transfer processing speed: the faster the speed, the lower the cost. The equipment depreciation for transfer and other equipment is calculated by dividing the equipment prices by the expected equipment lifetime of 60 months. Maintenance and material costs contain the prices for factory space, labor, and raw materials.

Comment 4: *Sometimes subjective words are added in the text like “outstanding” which is better to be avoided and replaced with scientific data. Also sometimes the word “we” is used, whereby the passive voice would be advised.*

Our response: We appreciate the reviewer’s helpful comment regarding inappropriate subjective words in the manuscript. Adopting the reviewer’s comment, we excluded or replaced some subjective words such as “outstanding”, “ultrahigh”, and “ultrasml” with scientific data. Also, passive voice sentences were used instead of the word “we”.

Modification to the manuscript:

We modified the manuscript to replace some subjective words with scientific data.

- (1) “Finally, flexible μ LEDs and transistors are fabricated via μ VAST with uniform electrical/optical properties and excellent mechanical stability (less than 9 % performance degradation).” on page 2 of revised manuscript.
- (2) “Herein, micro-vacuum assisted selective transfer (μ VAST) of inorganic thin-film semiconductors was introduced for realizing high-density soft electronics on unusual substrates.” on page 5 of revised manuscript.
- (3) “Finally, a high-performance flexible μ LED device was demonstrated on a polyimide (PI) substrate with an average transfer yield of 98.06 %, showing uniform optical power intensity and excellent mechanical stability.” on page 6 of revised manuscript.
- (4) “This structural LAZ modification enabled the fabrication of 20 μ m-sized holes with a high aspect ratio (>5) on the glass substrate.” on page 9 of revised manuscript.
- (5) “In summary, μ VAST technology was developed to transfer-print inorganic semiconductor arrays onto unusual substrates by controlling the micro-vacuum suction

force.” on page 17 of revised manuscript.

- (6) “Currently, we are developing the LIE-based VCM with an ultrahigh aspect ratio (>15) μ -holes on a quartz substrate for the large-area mass transfer printing of microchips smaller than $20 \mu\text{m}$. Furthermore, the μ VAST of RGB full-color μ LEDs with a flip-chip structure is being investigated for the commercialization of flexible μ LED applications. The μ VAST can provide essential breakthroughs for realizing a wide range of high-performance inorganic soft electronics.” on pages 17-18 of revised manuscript.

Comment 5: *It would be good to clearly add the limitations of the current technique (from materials and geometrical perspectives). For example, what are the smallest structures that can be manufactured and transferred and what is the percentage error in alignment?)*

Our response: We thank the reviewer’s valuable comment regarding the limitations of the μ VAST technique. As the reviewer’s opinion, we present μ VAST limitations in terms of the minimum transfer size and alignment error.

1. Minimum size of transferable microchips

In the case of current μ VAST, the minimum size of transferable microchips is $\sim 60 \mu\text{m}$. The size limitation of transferable microchips is related to the inner diameter of the μ -pillar. Since the inner diameter of SU-8 μ -pillar cannot be reduced under $60 \mu\text{m}$ in our photolithography process on LIE-drilled glass, only the microchips with a size over $60 \mu\text{m}$ can be picked-up without air leakage between μ LED and μ -pillar. In order to overcome this size limitation, we are developing a new fabrication method for smaller μ -holes and μ -pillars. Advanced LIE technology on thick quartz ($\sim 1 \text{ mm}$ thickness) is utilized to decrease the dimension of μ -holes and μ -pillars, which are simultaneously manufactured by subtractive etching of quartz substrate without the formation of additional SU-8 patterns.

2. Alignment error

The average alignment errors of μ VAST were 7.5 and $4.6 \mu\text{m}$ in the lateral and vertical directions, respectively, which were comparable to those of other transfer printing technologies (Supplementary Fig. S8). To minimize the alignment errors during the μ VAST process, we are upgrading our customized μ VAST equipment to include an auto-alignment system with an image sensing unit.

Adopting the reviewer’s comment, we modified the manuscript and Supplementary Information to

present the limitations of the current μ VAST process, and the research plans to overcome these issues.

Modification to the manuscript:

- (1) We added a SEM image in Supplementary Fig. S11 to show the minimum size of transferable microchips.

Fig. S11 | Transfer-printed microchips with various chip sizes. SEM image of 60/70/80 μm -sized microchips transferred on the flexible substrate. The minimum chip size of transferable microchips is 60 μm .

- (2) We revised the manuscript to present the alignment errors in μ VAST and the strategy to overcome this limitation.

“In order to minimize alignment errors during μ VAST, the auto-alignment system is being developed to upgrade the customized μ VAST equipment.” on pages 13-14 of revised manuscript.

- (3) We modified the conclusion to present the limitations of μ VAST and research plans to overcome these issues.

“Currently, we are developing the LIE-based VCM with an ultrahigh aspect ratio (>15) μ -holes on a quartz substrate for the large-area mass transfer printing of microchips smaller than 20 μm . Furthermore, the μ VAST of RGB full-color μ LEDs with a flip-chip structure is being investigated for the commercialization of flexible μ LED applications. The μ VAST can provide essential breakthroughs for realizing a wide range of high-performance inorganic soft electronics.” on pages 17-18 of revised manuscript.

Comment 6: *Certain bridge widths were chosen and studied, what is the max bridge width that can be used provided the maximum suction force of the micro pump?*

Our response: We appreciate the reviewer’s valuable comment regarding the maximum bridge width at maximum suction force. According to the finite element method (FEM) simulation results, the maximum principal stress of 669 MPa was applied at the end of the μ -bridge during the pick-up process. In this case, the fracture forces of each μ -bridge were calculated as 157, 184, and 212 μ N for bridge widths of 1.00, 1.25, and 1.50 μ m, respectively. The fracture force of μ -bridge proportionally increased with bridge width. Based on these results, the maximum bridge width is extrapolated to 2.17 μ m, which can be transferred via a micro-vacuum suction force of 286 μ N. Adopting the reviewer’s comment, we modified the Supplementary Information to present the maximum μ -bridge width that can be transferred via the μ VAST process.

Modification to the manuscript:

We added Supplementary Fig. S12 to present the maximum μ -bridge width for transfer printing of thin-film microchips.

Fig. S12 | Maximum μ -bridge width for μ VAST process. By extrapolating the relation between the fracture force and bridge width, the maximum bridge width is 2.17 μ m for transfer printing via a micro-vacuum suction force of 286 μ N.

Comment 7: *Regarding controlling the suction force, can you please provide more details about that and the range of forces that can be applied?*

Our response: We appreciate the reviewer’s helpful comment regarding controlling the suction force and range of adhesion force. In order to control the pressure inside the μ -channel, the

VCM was directly connected to an external vacuum pump through a vacuum hose. The suction force of μ VAST could be simply modulated by depressurizing and venting the μ -channel.

During the pick-up process, the micro-vacuum suction force was irrelevant to the μ -channel shape (e.g. channel width, length, and height) because the pressure inside the μ -channel remained constant regardless of the channel shape. Instead, the suction force was directly related to the size of the suction hole, which proportionally increased with the suction hole size. As mentioned in *Comment 6 of the Reviewer 1*, a suction force of 286 μ N was generated at 10×10 μ -hole arrays to pick-up the microchips because the pressure inside the μ -channel decreased to 316 mTorr during the depressurization state. We experimentally confirmed that a suction force of 286 μ N was enough to pick-up the microchips with μ -bridge widths of 1.00, 1.25, and 1.50 μ m.

During the printing procedure, the suction force was removed by venting the μ -channel to release the microchip arrays onto the target substrate. In this step, only the van der Waals force of 85 pN was applied between the μ -pillar and μ LED, holding μ LED arrays on VCM without vacuum suction force. Therefore, the adhesion force could be modulated from 85 pN to 286 μ N by adjusting the pressure inside the μ -channel. Adopting the reviewer's comment, we modified the Supplementary Information to show the range of suction force in the μ VAST process.

Modification to the manuscript:

We added Supplementary Fig. S13 to show the range of adhesion force for the μ VAST process.

Fig. S13 | Range of adhesion force for μ VAST. The adhesion force from 85 pN to 286 μ N is applied on microchips by controlling the micro-vacuum suction force.

Reviewer #2:

Overall comment: *In this manuscript, the authors introduced a micro-vacuum assisted selective transfer (μ VAST) approach for releasing and transferring micro-sized inorganic thin-film semiconductor devices such as micro-LEDs, transistors, etc. The topic is worth investigating and the authors have reported some important results. Specifically, due to the high adhesion switchability of the presented method, the author successfully transferred some embodiments. This manuscript can be re-evaluated for publication provided the authors sufficiently address the following minor comments.*

Our response: We sincerely appreciate the reviewer for the high evaluation of our research.

Comment 1: *This approach only presented the release/transfer of planar or relatively flat devices with smooth surfaces. In my opinion, some non-planar geometries of the semiconductor chips will lead to the inapplicability of the proposed transfer printing process. Can authors provide examples of transferring objects which have different roughness, 3D structures, or curved surfaces to demonstrate that this approach is universal?*

Our response: We appreciate the reviewer's helpful comment regarding the universal transfer printing of non-planar objects. In order to demonstrate the versatility of μ VAST in non-planar materials, we conducted transfer printing of micro-sized quartz beads (BCR130, Sigma-Aldrich) with particle sizes ranging from 50 to 220 μm . The μ -bead arrays with different surface roughness and morphologies were picked-up and released on the polyimide (PI) substrate via the μ VAST process. Furthermore, despite of 2 μm step height of the polymer pattern, the polymer-patterned silicon arrays (Figure 4f) were transferred on the PI film by micro-vacuum suction force, which was enabled by the sealing effect of a μ -pillar. Adopting the reviewer's comment, we modified the manuscript and Supplementary Information to verify the universality of μ VAST in the transfer printing of non-planar objects.

Modification to the manuscript:

- (1) We added Supplementary Fig. S9 to show the transfer printing of non-planar objects via the μ VAST process.

Fig. S9 | Transfer printing of non-planar objects via μ VAST. a, SEM image of transfer-printed μ -bead arrays on a PI substrate. b, Magnified SEM image of a transferred μ -bead.

(2) We revised the manuscript to present the universality of μ VAST in non-planar objects.

“Furthermore, not only the flat semiconductor microchips, non-planar objects with rough surfaces were transfer-printed through μ VAST, as shown in Supplementary Fig. S9.” on page 14 of revised manuscript.

Comment 2: *Because the fracture force of μ -bridge measured/simulated by the authors highly varied with the width change ranging from 1 to 1.5 μ m, the geometrical parameters of μ -bridge seem very critical to successfully transferring the semiconductor devices. However, the design or number of these anchors or bridges may vary depending on the lateral dimensions of semiconductor devices. Could the authors provide the required minimum vacuum forces according to more broad width range of μ -bridge (e.g., 1 to less than 30 μ m with a certain fixed thickness) without any collapse of the h-PDMS μ -channel?*

Our response: We thank the reviewer’s valuable comment regarding the required minimum vacuum force for a broader range of μ -bridge width. The minimum vacuum suction force to break the μ -bridge proportionally increased with the bridge width. By extrapolating the fracture forces of μ -bridges with the widths of 1.00, 1.25, and 1.50 μ m, the relation between the μ -bridge width and minimum suction force for μ VAST could be expressed as below.

$$(\text{Required minimum vacuum force}) = 109.7 \times (\mu\text{-bridge width}) + 47$$

This comment is similar to our response and modifications to *Comment 6 of the Reviewer 1*. Adopting the reviewer’s comment, we modified the Supplementary Information to present the required minimum vacuum force for a broader range of μ -bridge widths.

Modification to the manuscript:

As the reviewer's opinion, we modified the Supplementary Information with additional data. This comment is similar to our response and modifications to *Comment 6 of the Reviewer 1*. Please refer to our response and modifications to *Comment 6 of the Reviewer 1* related to the required suction force for a broader range of μ -bridge width.

Comment 3: *The fracture forces shown in Figures 3d and 3f do not agree with each other.*

Our response: We appreciate the reviewer's helpful comment regarding the different fracture forces between the nano-indentation measurement (Fig. 3d) and pick-up simulation (Fig. 3f).

Figure 3d shows the required indentation force to break the μ -bridge depending on the bridge width. In the case of the nano-indentation test, the end of the indenter tip pushed down a μ LED until the bridge fracture. As the indenter applied indentation force at the center point of a μ LED, the μ -bridge bent in a downward direction like a cantilever beam. During this downward bending of the μ -bridge, the maximum stress was concentrated at the upper end of the μ -bridge.

On the other hand, Figure 3f displays the calculated μ -bridge fracture forces depending on the bridge width. During the pick-up process, a μ LED was lifted-up by the vacuum suction force, which was applied to the circular area (inner circle of μ -pillar) of a μ LED center. In the FEM fracture simulation, we assumed that no physical deformation occurred in this circular suction region. Additionally, the μ -bridge was deformed to have an 'S' shape in an upward direction during the pick-up process. This S-shaped deformation induced stress concentrations at both ends of the bridge.

Consequently, the variations in stress distributions of μ -bridge caused different fracture forces between the nano-indentation measurement and pick-up simulation. Adopting the reviewer's comment, we modified the manuscript and Supplementary Information to present the different stress distributions in μ -bridge during the nano-indentation and pick-up process.

Modification to the manuscript:

- (1) We modified the manuscript to explain the reason for the different fracture forces between the nano-indentation measurement and pick-up simulation.

“The little difference in fracture force between the nano-indentation measurement and pick-up simulation was derived from the different fracture behavior and stress distribution of each case (Supplementary Fig. S6). Nevertheless, these results confirmed that the micro-vacuum suction force was large enough to break the μ -bridge for reliable transfer printing.” on page 12 of revised manuscript.

- (2) We added Supplementary Fig. S6 to present the stress distribution differences between the nano-indentation and pick-up process.

Fig. S6 | FEM simulation of stress distribution in μ -bridge. a, Stress distribution in μ -bridge during the nano-indentation. **b,** Stress distribution in μ -bridge during the pick-up process.

- (3) We revised Figures 3d and 3f to display the fracture behavior during the nano-indentation and pick-up process, respectively.

Fig. 3 | Mechanism analysis of μ VAST. d, Nano-indentation results of μ -bridges to experimentally investigate the fracture forces. The inset displays the fracture behavior during the nano-indentation. **f,** Comparison of suction force and fracture force of μ -bridge depending on the bridge width. The inset shows the fracture behavior during the pick-up process.

Comment 4: Line 223: The authors stated, “The maximum principal stress of 669 MPa was

concentrated at the end of the μ -bridge for fracture.” However, compared with Figure 1c inset, and Figure 4d,e, the fracture locations are not at the junction. How would the authors explain this observation based on their finite element simulation?

Our response: We appreciate the reviewer’s valuable comment regarding the fracture location of μ -bridge. In the FEM simulation of μ -bridge fracture, we assumed the μ -bridge with an ideal straight shape. In this case, the maximum principal stress was concentrated at the end of the μ -bridge. However, the real μ -bridge had a randomly serpentine shape after the sequential dry and wet etching processes. The serpentine-shaped μ -bridge had non-uniform widths at each point, showing different fracture behavior compared to a straight μ -bridge with uniform width distribution. According to the FEM simulation of stress distribution in serpentine μ -bridge, the maximum principal stress was concentrated at the narrowest point (a kind of notch) of the μ -bridge, inducing fracture in the middle of the μ -bridge. Adopting the reviewer’s comment, we modified the Supplementary Information to show the fracture behavior in the serpentine-shaped μ -bridge.

Modification to the manuscript:

We added Supplementary Fig. S14 to present the fracture behavior in the serpentine μ -bridge.

Fig. S14 | Fracture behavior in the serpentine μ -bridge. a, Magnified SEM image of the serpentine-shaped μ -bridge. **b,** FEM simulation of fracture behavior in serpentine-shaped μ -bridge.

Comment 5: *The adhesion switchability of this presented approach is quite high. Can the authors mention the range of the micro-vacuum suction force and how they calculated the value in the main text, instead of just in the supplementary note?*

Our response: We appreciate the reviewer’s remark about the calculation method of adhesion switchability. As the reviewer suggested, we added the calculation details of adhesion switchability to the main text of the manuscript.

Modification to the manuscript:

- (1) We revised the manuscript to provide the calculation details of adhesion switchability.

“In the case of μ VAST, the adhesion switchability was the ratio of maximum adhesion force to minimum adhesion force, which was determined by the following equation (1)²⁰.

$$\text{Adhesion switchability} = \frac{F_{max}}{F_{min}} = \frac{F_{suction} + F_{vdW}}{F_{vdW}} \quad (1)$$

where F_{max} is the maximum adhesion force, which is expressed as the sum of the micro-vacuum suction force and the van der Waals force. $F_{suction}$ is the vacuum suction force generated at the μ -holes and F_{vdW} is the van der Waals force between the microchips and the VCM. On the other hand, F_{min} is the minimum adhesion force, which is defined by only the van der Waals force without the vacuum suction force, because the vacuum suction force is removed by venting the μ -channel. F_{vdW} , the van der Waals interaction between two contact surfaces, could be calculated by equation (2)⁴⁶.

$$F_{vdW} = \frac{A}{6\pi D^3} \times \text{Contact Area} \quad (2)$$

where A is the Hamaker constant and D is the separation distance between the microchips and the VCM⁴⁶. Hamaker constant (A) is a physical coefficient to define the van der Waals interaction between two contact bodies⁴⁷. Separation distance (D) is measured by an atomic force microscopy (AFM) analysis because the separation distance of two contacted surfaces is determined by the roughness of each surface⁴⁶. According to the AFM results (Supplementary Fig. S4), a μ LED had a relatively large surface roughness value compared to that of the VCM. In this case, the separation distance was equivalent to the distance between a smooth surface and a rough surface, which was the maximal roughness peak (d_{max}) of a rough surface. Based on this approximation, the separation distance between the μ LED and the VCM was expressed as below.

$$D \approx d_{max} \text{ of } \mu\text{LED surface} \quad (3)$$

The contact area between the μ LED and the VCM (Supplementary Fig. S5) could be calculated as follows:

$$\text{Contact area} = \text{Area of } \mu\text{LED} - \text{Inner circle area of } \mu\text{-pillar} \quad (4)$$

Based on equations (2) ~ (4), the F_{vdW} of μ VAST was 85 pN. As the $F_{suction}$ of 286 μ N was generated by an air pressure difference between the inside and outside of the μ -channel during the pick-up process, the adhesion switchability of μ VAST could be calculated as 3.364×10^6 by equation (1). The adhesion switchability of μ VAST was three orders of magnitude higher compared to that of previously reported transfer methods, thus demonstrating the superior controllability and reliability of μ VAST (see Supplementary Note 1 for calculation details).” on pages 7-9 of revised manuscript.

- (2) We added a reference in the manuscript for a better understanding of the calculation method for adhesion switchability.

47. Visser, J. On Hamaker constants: A comparison between Hamaker constants and Lifshitz-Van der Waals constants. *Adv. Colloid Interface Sci.* 3, 331-363 (1972). on page 26 of revised

manuscript.

Comment 6: *Figure 3c: Can the authors transfer print an array of μ LEDs onto an uneven surface such as the hornet wing using μ VAST?*

Our response: We appreciate the reviewer's valuable comment regarding the transfer printing of microchip arrays onto uneven surfaces. As the reviewer suggested, we demonstrated the transfer print of μ LED arrays onto arbitrary substrates with a high surface roughness including fabric and leaf, as shown in revised Fig. 4c. During the release process of μ VAST, the μ LED arrays conformally contacted with a rough surface, followed by venting the μ -channel to print the microchip arrays. Due to the ultrahigh adhesion switchability of μ VAST, μ LED arrays were released onto uneven surfaces via van der Waals interactions between μ LEDs and the target surface without additional adhesives. Adopting the reviewer's comment, we modified the manuscript to demonstrate the transfer printing of microchip arrays onto uneven surfaces.

Modification to the manuscript:

- (1) We added SEM images in Fig. 4c to demonstrate the transfer printing of microchip arrays onto uneven surfaces.

Fig. 4 | Universal transfer printing of thin-film semiconductors via μ VAST. c, SEM images of transferred μ LEDs on a hornet wing, leaf, and fabric.

(2) We revised the manuscript to present the μ VAST of thin-film microchip arrays on uneven substrates.

“In addition, the μ LED arrays were successfully transfer-printed onto arbitrary substrates such as human skin, non-sticky rubber, paper, hornet wing, leaf, and even a fabric regardless of the surface adhesion forces and morphologies of the target substrates, as shown in Fig. 4b and 4c.” on page 14 of revised manuscript.

Comment 7: *Figure 5c: Are the μ LEDs individually controllable?*

Our response: We appreciate the reviewer’s helpful comment regarding the individual control of μ LED arrays. The vertical-structured 10×10 thin-film μ LEDs were transfer-printed on ACF laminated bottom electrode (BE), followed by a thermo-compressive bonding for electrical interconnection. Then, transferred μ LEDs were electrically isolated from each other via a parylene-C polymer layer, followed by top electrode (TE) formation on the top surface of μ LEDs. In order to individually control 10×10 μ LEDs, the transferred μ LEDs were connected to 10 bottom electrodes and 10 top electrodes with a crossbar structure. Finally, each μ LED could be individually driven by sweeping a voltage applied on the top electrode (set bottom electrode as a ground state) with a forward voltage of 7.4 V. To clearly show the individual controllability of the transferred μ LEDs, we added optical images and I-V curve in the Supplementary Information.

Modification to the manuscript:

We added Supplementary Fig. S15 to demonstrate the individual control of 10×10 transferred μ LEDs.

Fig. S15 | Individual driving of transferred μ LEDs. **a**, Optical image of 10×10 flexible μ LEDs with 10 bottom electrodes and 10 top electrodes. **b**, Optical image of an individually driven μ LED. The inset shows the magnified OM image of an operating μ LED. **c**, I-V characteristic of an individually driven μ LED.

Comment 8: *The Outlook section seems more like the conclusion. Please include more discussions on the outlook, challenges, and opportunities.*

Our response: We appreciate the reviewer's valuable comment regarding the inclusion of discussion in the outlook section. We agreed with the reviewer's opinion to provide challenges and opportunities of current μ VAST as below.

Modification to the manuscript:

- (1) We modified the manuscript to provide challenges and opportunities of current μ VAST.

“Currently, we are developing the LIE-based VCM with an ultrahigh aspect ratio (>15) μ -holes on a quartz substrate for the large-area mass transfer printing of microchips smaller than 20 μ m. Furthermore, the μ VAST of RGB full-color μ LEDs with a flip-chip structure is being investigated for the commercialization of flexible μ LED applications. The μ VAST can provide essential breakthroughs for realizing a wide range of high-performance inorganic soft electronics.” on pages 17-18 of revised manuscript.

- (2) We changed the “Outlook” session into the “Discussion” session.

Reviewer #3:

Overall comment: *The paper reported a design of active stamp, which consists of PDMS micro-channels, glass micro-hole arrays, and SU-8 sealing rings around the micro-holes, to control the pick-up and release of microchips through air pressure. It is shown that the stamp offers a large adhesion switchability, which enables the successful transfer printing of microchips with diverse materials and dimensions onto various unconventional substrates such as human skin, rubber, paper, etc. The work is solid and represents a good advance in the field of transfer printing. Several major issues should be addressed before its consideration for publication.*

Our response: We deeply appreciate the reviewer's high evaluation of our manuscript.

Comment 1: *Universal in the title is not appropriate since the proposed approach is only good for flat objects.*

Our response: We thank the reviewer's helpful comment regarding the title of universal transfer. This comment is related to our response and modifications to *Comment 1 of the Reviewer 2*. Adopting the reviewer's comment, we transfer-printed the non-planar objects (micro-sized beads) with rough surfaces to verify the universality of μ VAST. Now, we believe that the title of universal transfer is appropriate for the μ VAST.

Modification to the manuscript:

As the reviewer's opinion, we modified the Supplementary information with additional data. This comment is similar to our response and modifications to *Comment 1 of the Reviewer 2*. Please refer to our response and modifications to *Comment 1 of the Reviewer 2* related to the transfer printing of non-planar objects.

Comment 2: *The reported adhesion switchability is not convincing. The definition of adhesion switchability should be the ratio of the maximum adhesion strength (or force) to the minimum adhesion strength (or force). The suction force could be positive or negative depending on the air pressure, which may yield an infinite adhesion switchability. The reported adhesion switchability in the manuscript is not appropriate and the related part should be revised.*

Our response: We appreciate the reviewer's helpful comment regarding the calculation method for adhesion switchability. We agreed with the reviewer's suggestion that adhesion switchability was the ratio of maximum adhesion force to minimum adhesion force. In the case of μ VAST, the maximum adhesion force was applied on a microchip during the pick-up process, which was the sum of the vacuum suction force and the van der Waals force between the microchip and the SU-8 μ -pillar.

On the other hand, minimum adhesion was expressed as just the van der Waals force between the microchip and the μ -pillar, because vacuum suction force was removed by venting the μ -channel during the release process. Therefore, the adhesion switchability of μ VAST could be defined as the ratio of the sum of the van der Waals force and the vacuum suction force to only the van der Waals force, which was identical to the ratio of maximum adhesion force to minimum adhesion force. To clearly explain the calculation method, we added the equation and the description of the adhesion switchability in the manuscript and Supplementary Information.

Modification to the manuscript:

We modified the manuscript and Supplementary information to clearly explain the calculation method of adhesion switchability.

- (1) “In the case of μ VAST, the adhesion switchability was the ratio of maximum adhesion force to minimum adhesion force, which was determined by the following equation (1)²⁰.

$$\text{Adhesion switchability} = \frac{F_{max}}{F_{min}} = \frac{F_{suction} + F_{vdW}}{F_{vdW}} \quad (1)$$

where F_{max} is the maximum adhesion force, which is expressed as the sum of the micro-vacuum suction force and the van der Waals force. $F_{suction}$ is the vacuum suction force generated at the μ -holes and F_{vdW} is the van der Waals force between the microchips and the VCM. On the other hand, F_{min} is the minimum adhesion force, which could be defined by only the van der Waals force without the vacuum suction force, because the vacuum suction force is removed by venting the μ -channel.” on pages 7-8 of revised manuscript.

- (2) “The adhesion switchability of μ VAST is the ratio of maximum adhesion force to minimum adhesion force, which is determined by the following equation (1).

$$\text{Adhesion switchability} = \frac{F_{max}}{F_{min}} = \frac{F_{suction} + F_{vdW}}{F_{vdW}} \quad (1)$$

where F_{max} is the maximum adhesion force, which can be expressed as the sum of the micro-vacuum suction force and the van der Waals force. $F_{suction}$ is the vacuum suction force generated at the μ -holes and F_{vdW} is the van der Waals force between the microchips and the SU-8 μ -pillar. F_{min} is the minimum adhesion force, which can be defined by only the van der Waals force, because the vacuum suction force is removed by venting the μ -channel.” on page 4 of revised Supplementary Information.

Comment 3: *What are the factors related to the minimum pitch of μ -hole arrays? Is it possible to transfer devices with a small pitch?*

Our response: We appreciate the reviewer's valuable comment regarding the minimum pitch of μ -hole arrays and transferred devices. The minimum pitch of μ -hole arrays is directly related to the distance between each laser shot in the LIE process. The distance between 20 μm -sized μ -hole arrays could be reduced to less than 5 μm by optimizing the laser frequency and chemical etching time.

In the case of the μ VAST process of freestanding μ LEDs, the minimum pitch of μ -hole arrays should be determined by considering the pitch of freestanding μ LEDs. During the chemical etching process for freestanding μ LEDs, the underlying GaAs substrate was anisotropically etched in both lateral and vertical directions with a lateral dimension of about 150 μm . Therefore, the minimum pitch of μ LEDs was 300 μm by considering the width of lateral etching and supporting parts at both sides in order to ensure a stable structure of the freestanding μ LED arrays after the etching process. Accepting the reviewer's comment, we revised Supplementary Information to present the factor related to the minimum pitch of transferred devices.

Modification to the manuscript:

We added Supplementary Fig. S16 to present the factor related to the minimum pitch of transferred devices.

Fig. S16 | Factor related to the pitch of μ LED arrays. a, Cross-section SEM image of the freestanding μ LED. **b,** μ LED arrays with a lateral pitch of 300 μm .

Comment 4: *Can the size of the transferred object be further reduced? What is the limited size of the transferred object?*

Our response: We thank the reviewer's remark about the minimum size of microchips that can be transferred. The size of microchips can be reduced to 60 μm for reliable transfer printing without air leakage. This comment is similar to our response and modifications to *Comment 5 of the Reviewer 1*. Please refer to our response and modifications to *Comment 5 of the Reviewer 1* related to the minimum chip size of transferable microchips.

Modification to the manuscript:

As the reviewer's opinion, we modified the Supplementary information with additional data. This comment is similar to our response and modifications to *Comment 5 of the Reviewer 1*. Please refer to our response and modifications to *Comment 5 of the Reviewer 1* related to the minimum chip size of transferable microchips.

Comment 5: *How many μ LEDs are involved in a single test in the transfer yield repeatability test?*

Our response: We thank the reviewer's remark about the number of μ LEDs in a single transfer cycle. The 10×10 μ LED arrays (100 μ LEDs) were transfer-printed in a single test. We repeated this transfer cycles 100 times to evaluate the average transfer yield of μ VAST.

Modification to the manuscript:

We modified the manuscript to clearly present the number of μ LEDs in a single transfer test.

“Figure 3j shows the average transfer yield and suction force during 100 μ VAST cycles of 10×10 μ LED arrays.” on page 13 of revised manuscript.

Comment 6: *It is hard to form tightly conformal contact between the hard SU-8 sealing rings and chips. It is better to study whether the sealing ring exists or not on the sealing effect.*

Our response: We appreciate the reviewer's helpful comment regarding the sealing effect of SU-8 μ -pillar. The μ -pillar arrays enhanced the vacuum suction force by minimizing air leakage caused by contaminations or defects on the μ LED surface. To demonstrate the sealing effect of SU-8 μ -pillar, we carried out the μ VAST process without μ -pillar arrays. The transfer yield without SU-8 μ -pillar was 80 % (14 missing chips and 6 unwanted contamination) due to air leakage at the interface between μ LEDs and VCM. Furthermore, the transferred μ LEDs exhibited a large misalignment because the VCM could not tightly hold μ LED arrays due to the decreased vacuum suction force. Based on these reasons, the μ -pillar structure is critical for the μ VAST process with high transfer yield and positional accuracy. Adopting the reviewer's comment, we revised Supplementary Information to exhibit the sealing effect of SU-8 μ -pillar.

Modification to the manuscript:

We added Supplementary Fig. S17 to present the sealing effect of SU-8 μ -pillar during the μ VAST process.

Fig. S17 | 10×10 μ LEDs transferred via μ VAST without μ -pillar arrays. The transfer yield of non-pillar μ VAST is 80 % with 14 missing chips and 6 contaminations.

REVIEWERS' COMMENTS

Reviewer #2 (Remarks to the Author):

I feel that the authors have addressed all of the comments and suggestions from the referees. The revised version is now suitable for publication.

Reviewer #3 (Remarks to the Author):

The authors addressed this reviewer's comments satisfactorily and the paper can be published as it is.

[Response to the reviewers]

Reviewer #2:

Overall Comment: *I feel that the authors have addressed all of the comments and suggestions from the referees. The revised version is now suitable for publication.*

Our response: We sincerely appreciate the reviewer's high evaluation of our work.

Reviewer #3:

Overall comment: *The authors addressed this reviewer's comments satisfactorily and the paper can be published as it is.*

Our response: We sincerely appreciate the reviewer's high evaluation of our research.